# Learning Generalized Gumbel-max Causal Mechanisms

**Guy Lorberbom**[*]
Technion
Haifa, Israel
guy_lorber@campus.technion.ac.il

**Daniel D. Johnson**[*]
Google Research
Toronto, ON, Canada
ddjohnson@google.com

**Chris J. Maddison**
University of Toronto
& Vector Institute
Toronto, ON, Canada
cmaddis@cs.toronto.edu

**Daniel Tarlow**
Google Research
Montreal, QC, Canada
dtarlow@google.com

**Tamir Hazan**
Technion
Haifa, Israel
tamir.hazan@technion.ac.il

## Abstract

To perform counterfactual reasoning in Structural Causal Models (SCMs), one needs to know the causal mechanisms, which provide factorizations of conditional distributions into noise sources and deterministic functions mapping realizations of noise to samples. Unfortunately, the causal mechanism is not uniquely identified by data that can be gathered by observing and interacting with the world, so there remains the question of how to choose causal mechanisms. In recent work, Oberst & Sontag (2019) propose Gumbel-max SCMs, which use Gumbel-max reparameterizations as the causal mechanism due to an intuitively appealing *counterfactual stability* property. In this work, we instead argue for choosing a causal mechanism that is best under a quantitative criteria such as minimizing variance when estimating counterfactual treatment effects. We propose a parameterized family of causal mechanisms that generalize Gumbel-max. We show that they can be trained to minimize counterfactual effect variance and other losses on a distribution of queries of interest, yielding lower variance estimates of counterfactual treatment effect than fixed alternatives, also generalizing to queries not seen at training time.

## 1 Introduction

Pearl [2009] presents a "ladder of causation" that distinguishes three levels of causal concepts: associational (level 1), interventional (level 2), and counterfactual (level 3). As an illustrative example, suppose we wish to compare two treatments for a patient in a hospital. Level 1 corresponds to information learnable from passive observation, e.g. correlations between treatments given in the past and their outcomes. Level 2 coresponds to active intervention, e.g. choosing which treatment to give to a new patient, and measuring the distribution of outcomes it causes (called an *interventional distribution*). Level 3 corresponds to reasoning about hypothetical interventions given that some other outcome actually occurred (called a *counterfactual distribution*): given that the patient recovered after receiving a specific treatment, what would have happened if they received a different one? The three levels are distinct in the sense that it is generally not possible to uniquely determine higher level models from lower level information [Bareinboim et al., 2020]. In particular, although we can determine level 2 information (such as the average effect of each treatment) by actively intervening

---

[*]Equal contribution

35th Conference on Neural Information Processing Systems (NeurIPS 2021).

in the world, we cannot determine level 3 information in this way (such as the effect two different treatments would have had for a single situation).

Nevertheless, we still desire to reason about counterfactuals. First, counterfactual reasoning is fundamental to human cognition, e.g., in assigning credit or blame, and as a mechanism for assessing potential alternative past behaviors in order to update policies governing future behavior. Second, counterfactual reasoning is computationally useful, for example allowing us to shift measurements from an observed policy to an alternative policy in an off-policy manner [Buesing et al., 2018, Oberst and Sontag, 2019].

Doing counterfactual reasoning thus requires us to make an assumption about the *causal mechanism* of the world, which specifies how particular choices lead to particular outcomes while holding "everything else" fixed. Different assumptions, however, lead to different counterfactual distributions. One approach, as exemplified by Oberst and Sontag [2019], is axiomatic. There, an intuitive requirement of counterfactuals is presented, and then a causal mechanism is chosen that provably satisfies the requirement. The resulting proposal is *Gumbel-max SCMs* which assume causal mechanisms are governed by the Gumbel-max trick.

In this work, we instead view the choice of causal mechanism as an optimization problem, and ask what causal mechanism (that is consistent with level 2 observations) yields the most desirable behavior under some statistical or computational criterion. For example, what mechanism leads to the lowest variance estimates of treatment effects in a counterfactual setting? If we are ultimately interested in estimating a level 2 property, choosing a level 3 mechanism that minimizes the variance of our estimate can lead to algorithms that converge faster. Alternatively, we can view minimizing variance as a kind of stability assumption on the treatment effect: we are interested in the causal mechanism for which the treatment effect is as "evenly divided" as possible across realizations of the exogenous noise. More generally, casting the problem in terms of optimization gives additional flexibility to choose a causal mechanism that is *specifically-tuned* to a distribution of observations and interventions of interest, and in terms of a loss function that measures the quality of a counterfactual sample. A key insight to lay the foundation for specifically-tuned causal mechanisms is to view the average treatment effects (or other measure of interest) from the perspective of a coupling between interventional and counterfactual distributions.

We begin by drawing connections between causal mechanisms and couplings, and show that defining a level 3 structural causal model consistent with level 2 observations is equivalent to defining an *implicit coupling* between interventional distributions. Next, to motivate the need for specifically-tuned causal mechanisms, we prove limitations of non-tuned mechanisms (including Gumbel-max) and the power of tuned mechanisms by drawing on connections to literature on couplings, optimal transport, and common random numbers. We then introduce a continuously parameterized family of causal mechanisms whose members are identical when used in a level 2 context but different when used in a level 3 context. The families contain Gumbel-max, but a wide variety of other mechanisms can be learned by using gradient-based optimization over the family of mechanisms. Empirically we show that the mechanisms can be learned using a variant of Gumbel-softmax relaxation [Maddison et al., 2017, Jang et al., 2017], and that the resulting mechanisms improve over Gumbel-max and other fixed mechanisms. Further, we show that the learned mechanisms generalize, in the sense that we can learn a causal mechanism from a *training set* of observed outcomes and counterfactual queries and have it generalize to a *test set* of observed outcomes and counterfactual queries that were not seen at training time.

## 2 Background

**Structural Causal Models.** Here we briefly summarize the framework of structural causal models (SCMs), which enable us to ask and answer counterfactual questions. SCMs divide variables into *exogenous* background (or *noise*) variables and *endogenous* variables that are modeled explicitly. Each endogenous variable $v_i$ has a set of parent endogenous variables $pa_i$, an associated exogenous variable $u_i$, and a function $f_i$. Intuitively, the function $f_i$ specifies how the value of $v_i$ depends on the other variables $pa_i$ of interest, and $u_i$ represents "everything else" that may influence the value of $v_i$. Putting these together along with a prior distribution over $u_i$'s defines the *causal mechanism* governing $v_i$, as $v_i = f_i(pa_i, u_i)$; we emphasize that this mapping is deterministic, as all of the randomness in $v_i$ is captured by either $pa_i$ or $u_i$.

Marginalizing over the exogenous variable $u_i$ yields the interventional distribution $p(v_i|pa_i)$, which specifies how $v_i$ is affected by modifying its parents. Counterfactual reasoning, on the other hand, amounts to holding $u_i$ fixed but considering multiple values for $pa_i$ and consequently for $v_i$. In other words, if $v_i^{(1)} = f_i(pa_i^{(1)}, u_i)$ and $v_i^{(2)} = f_i(pa_i^{(2)}, u_i)$, the counterfactual distribution is $p(v_i^{(2)}|pa_i^{(2)}, v_i^{(1)}, pa_i^{(1)}) = \int_{u_i} p(v_i^{(2)}|pa_i^{(2)}, u_i)p(u_i|v_i^{(1)}, pa_i^{(1)})$.

**Gumbel-max SCMs**   The Gumbel-max trick is a method for drawing a sample from a categorical distribution defined by logits $l \in \mathbb{R}^K$, e.g. $p(X = k) \propto \exp l_k$. It is based on the fact that, if $\gamma_k \sim \text{Gumbel}(0)$ for $k \in \{1, \ldots, K\}$, then[2]

$$\mathbb{P}_\gamma \Big[ x = \text{argmax}_k \left[ l_k + \gamma_k \right] \Big] = \frac{\exp(l_x)}{\sum_k \exp(l_k)}. \tag{1}$$

Oberst and Sontag [2019] use this as the basis for their Gumbel-max SCM, which uses vectors of Gumbel random variables $\gamma \in \mathbb{R}^K$ as the exogenous noise, and specifies the causal mechanism for each variable $x$ with parents $pa$ as

$$x = f(pa, \gamma) = \text{argmax}_k \left[ l_k + \gamma_k \right], \tag{2}$$

where $l_k = \log p_k = \log p(X = k|pa)$ is the interventional distribution of $x$ given the choice of $pa$.

Oberst and Sontag [2019] motivate their Gumbel-max SCMs by appealing to a property known as *counterfactual stability*: if we observe $X^{(1)} = i$ under intervention $p^{(1)} = p$, then the only way we can observe $X^{(2)} = j \neq i$ under intervention $p^{(2)} = q$ is if the probability of $j$ relative to $i$ has increased, i.e. if $\frac{q_j}{p_j} > \frac{q_i}{p_i}$ or equivalently $\frac{q_j}{q_i} > \frac{p_j}{p_i}$. This property generalizes the well-known monotonicity assumption for binary random variables [Pearl, 1999] to the larger class of unordered categorical random variables, and corresponds to the intuitive idea that the outcome should only change in a counterfactual scenario if there is a reason for it to change (i.e., some other outcome's relative probability increased more than the observed outcome's). Oberst and Sontag [2019] show that Gumbel-max SCMs are counterfactually stable.

Gumbel-max SCMs have interesting properties due to the Gumbel-max trick introducing statistical independence between the max-value and the argmax, cf. Maddison et al. [2014]. Oberst and Sontag [2019] exploit this to sample from counterfactual distributions by sampling the exogenous variables $\gamma$ from the posterior over Gumbel noise: conditioned on an observed outcome $x = \text{argmax}_k \left[ l_k + \gamma_k \right]$ we can sample $\gamma_x \sim \text{Gumbel}(-l_x)$ and then sample the other $\gamma_i$ from truncated Gumbel distributions.

We observe that sharing the exogenous noise $\gamma$ between two different logit vectors $l^{(1)}$ and $l^{(2)}$ yields a joint distribution over pairs of outcomes:

$$\pi_{gm}(x, y) = \mathbb{P}_\gamma \Big[ x = \text{argmax}_k \left[ l_k^{(1)} + \gamma_k \right] \text{ and } y = \text{argmax}_k \left[ l_k^{(2)} + \gamma_k \right] \Big]. \tag{3}$$

We call this a *Gumbel-max coupling*.

**Couplings.**   A coupling between categorical distributions $p(x)$ and $q(y)$ is a joint distribution $\pi(x, y)$ such that

$$\sum_x \pi(x, y) = q(y) \text{ and } \sum_y \pi(x, y) = p(x). \tag{4}$$

The set of all couplings between $p$ and $q$ is written $C(p, q)$. A core problem of coupling theory is find a coupling $\pi \in C(p, q)$ in order to estimate $\mathbb{E}_{x \sim p}[h_1(x)] - \mathbb{E}_{y \sim q}[h_2(y)] = \mathbb{E}_{x,y \sim \pi}[h_1(x) - h_2(y)]$ for some real-valued cost functions $h_1, h_2$, with minimal variance $\text{Var}_{x,y \sim \pi}[h_1(x) - h_2(y)]$. Interestingly, whenever $h_1(x), h_2(y)$ are monotone, such a coupling can be attained by $(F_X^{-1}(U), F_Y^{-1}(U)))$, when $F_X(t), F_Y(t)$ are the cumulative distribution functions (CDFs) of $X, Y$ and $U$ is a uniform random variable over the unit interval. When $h_1, h_2$ are not monotone, there are clearly cases where CDF inversion produces suboptimal couplings. Couplings are also used to define the Wasserstein distance $W_1(p, q; d)$ between two distributions $p$ and $q$ (with respect to a metric $d$ between samples):

$$W_1(p, q; d) = \inf_{\pi \in C(p,q)} \mathbb{E}_{x,y \sim \pi} d(x, y), \tag{5}$$

When $d(x, y) = 1_{x \neq y}$, then a coupling that attains this infimum is known as a *maximal coupling*; such a coupling maximizes the probability that $X$ and $Y$ are the same.

---

[2]Samples of Gumbel(0) random variables can be generated as $-\log(-\log(u))$, where $u \sim \text{Uniform}([0, 1])$

**Causal mechanisms as implicit couplings.** Any causal mechanism for a variable $v_i$ defines a coupling between outcomes under two counterfactual interventions. In other words, for any two interventions $pa_i^{(1)}$ and $pa_i^{(2)}$ on the parent nodes $pa_i$, sharing the exogenous noise $u_i$ yields a coupling $\pi_{SCM}$ between the interventional distributions $p(v_i^{(1)}|do(pa_i^{(1)}))$ and $p(v_i^{(2)}|do(pa_i^{(2)}))$:

$$\pi_{SCM}\left(v_i^{(1)}, v_i^{(2)}\right) = \mathbb{P}_{u_i}\left[v_i^{(1)} = f_i(pa_i^{(1)}, u_i) \text{ and } v_i^{(2)} = f_i(pa_i^{(2)}, u_i)\right]. \tag{6}$$

We call this an *implicit coupling* because $f_i(\cdot, u)$ is not directly defined with respect to a particular pair of marginal distributions $p, q$, but instead arises from running the same causal mechanism forward with shared noise but different inputs, representing either $p$ or $q$.

This connection between SCMs and couplings enables us to translate ideas between the two domains. For instance, suppose we are interested in estimating $\mathbb{E}_{x \sim p}[h_1(x)] - \mathbb{E}_{y \sim q}[h_2(y)]$ between observed outcomes from $p \propto \exp l^{(1)}$ and counterfactual outcomes from $q \propto \exp l^{(2)}$. We might do so using counterfactual reasoning in the Gumbel-max SCM:

$$\mathbb{E}_{x,y \sim \pi}[h_1(x) - h_2(y)] = \sum_y q(y)\mathbb{E}_{\gamma \sim (G_2|y)}\left[h_1(\text{argmax}_k[l_k^{(1)} - l_k^{(2)} + \gamma_k]) - h_2(\text{argmax}_k[\gamma_k])\right].$$

Here $G_2|y$ is the Gumbel distribution with location $\log l_k^{(2)}$ conditioned on the event that $y$ is the maximal argument; the proof of this equality appears in Appendix B. However, if we are interested in minimizing the variance of a Monte Carlo estimate of the expectation, this may not be optimal.

# 3   Problem Statement: Building SCMs by Learning Implicit Couplings

In this section, we formalize the task of selecting causal mechanisms according to some quantitative metric. Recall our initial example of comparing two treatment policies for patients in a hospital. For simplicity, we consider a single action $a$ taken by a hypothetical treatment policy, which leads to a distribution over outcomes $v$. (More generally, we can let $a$ be the set of parents for any variable $v$ in a SCM.) As in Oberst and Sontag [2019], we assume we have access to all of the interventional distributions of interest without any latent confounders.

Our goal is to define a parameterized family of causal mechanisms consistent with the interventional distributions for all possible actions $a$. We assume that the set $V$ of possible outcomes is finite with $|V| = K$, but do not restrict the space of actions $a$; instead, we require that our causal mechanism can produce samples from any interventional distribution $p(v|do(a))$, expressed as a vector $l^{(a)} \in \mathbb{R}^K$ for which $p(v = k|do(a)) \propto \exp l_k^{(a)}$.

Specifically, let $\boldsymbol{u} \in \mathbb{R}^D$ be a sample from some noise distribution (e.g., from a Gumbel(0) distribution per dimension) and let $l \in \mathbb{R}^K$ be a vector of (conditional) logits defining a distribution over $K$ categorical outcomes. We wish to learn a function $f_\theta : \mathbb{R}^D \times \mathbb{R}^K \to \{1, \ldots, K\}$ that maps noise and logits to a sampled outcome. We require that the process produces samples from the distribution $p(k) \propto \exp l_k$ when integrating over $\boldsymbol{u}$ (i.e., we want a reparameterization trick), and also that we can counterfactually sample the exogenous noise $\boldsymbol{u}$ conditioned on an observation $x^{(obs)}$ (e.g. $\boldsymbol{u} \sim p(\boldsymbol{u}|l, f_\theta(\boldsymbol{u}, l) = x^{(obs)})$). We obtain an implicit coupling by running $f_\theta$ with the same noise and two different logit vectors $l^{(1)}$ and $l^{(2)}$. We can think of $l^{(1)}$ as the logits governing an observed outcome and $l^{(2)}$ as their modification under an intervention.

Each setting of the parameters $\theta$ produces a different SCM. We propose to learn $\theta$ in such a way as to approximately minimize an objective of interest. We provide two degrees of freedom for defining this objective. First, we must choose a loss function $g_{l^{(1)}, l^{(2)}} : \{1, \ldots, K\} \times \{1, \ldots, K\} \to \mathbb{R}$ that assigns a real-valued loss to a joint outcome $(f_\theta(\boldsymbol{u}, l^{(1)}), f_\theta(\boldsymbol{u}, l^{(2)}))$, perhaps modulated by $l^{(1)}$ and $l^{(2)}$. The loss function is used to specify how desirable a pair of observed and counterfactual outcomes are (e.g., if we are trying to minimize variance, the squared difference $(h(v^{(1)}) - h(v^{(2)}))^2$ of scores for each outcome). Second, we must choose a distribution $\mathcal{D}$ over pairs of logits $(l^{(1)}, l^{(2)})$. This determines the distribution of observed outcomes and counterfactual queries of interest.

Given these choices, our main objective is as follows:

$$\theta^* = \mathrm{argmin}_\theta \ \mathbb{E}_{(l^1,l^2)\sim\mathcal{D}}\mathbb{E}_{\boldsymbol{u}}\left[g_{l^{(1)},l^{(2)}}(f_\theta(\boldsymbol{u},l^{(1)}), f_\theta(\boldsymbol{u},l^{(2)}))\right] \tag{7}$$

$$\text{subject to } \ \mathbb{P}_{\boldsymbol{u}}[f_\theta(\boldsymbol{u},l) = k] = \frac{\exp l_k}{\sum_{k'}\exp l_{k'}} \qquad \text{for all } l \in \mathbb{R}^K. \tag{8}$$

**Relationship to 1-Wasserstein Metric.** We are free to set $g$ to be a distance metric $d$, in which case the implicit coupling between $f_\theta(\boldsymbol{u},l^{(1)})$ and $f_\theta(\boldsymbol{u},l^{(2)})$ bears similarity to the optimal 1-Wasserstein coupling for $d$. However, a key difference is that $f_\theta$ can be used to generate samples from one side of the coupling (say $p$) without knowledge of what $q$ will be chosen. Thus, $f_\theta$ can be seen as *coupling $p$ to all $q$ simultaneously*, in the same way that observing a particular outcome simultaneously induces counterfactual distributions for all alternative interventions. In contrast, the Wasserstein optimization requires knowledge of both $p$ and $q$ and then computes a coupling specific to that pair. We discuss the effect of this restriction in the next section.

**Interpretation in terms of causal inference.** To construct a full level 3 SCM, $f_\theta$ must be combined with a set of known level 2 interventional distributions $p(v|do(a))$, similar to the Gumbel-max SCM in this regard [Oberst and Sontag, 2019]. In particular $f_\theta$ and Gumbel-max assume there are no latent confounders, and that the set of outcomes is discrete. The objective $g$ and distribution $\mathcal{D}$ serve a similar role as counterfactual stability or monotonicity assumptions [Oberst and Sontag, 2019, Pearl, 1999], in that they are a-priori choices that select the intended level 3 mechanism from the set of consistent mechanisms. The main difference is that these assumptions are made at a higher level of abstraction. Instead of specifying the mechanism, we specify a family of mechanisms along with a quantitative quality measure that can be optimized.

## 4 Properties of Implicit Couplings

Our development so far raises questions about the relationship of the proposed approach to the Gumbel-max couplings underlying Gumbel-max SCMs and about the relationship of our objective to Wasserstein metrics. In this section we establish the relationships and differences. The main results are that despite being counterfactually stable, Gumbel-max couplings are not actually maximal couplings. We then go further and show that any *implicit coupling* is limited in expressivity. This establishes the difference to Wasserstein metrics and optimal transport, which are framed in terms of minimizing over the larger space of all couplings.

### 4.1 Non-maximality of Gumbel-max Couplings

We know that Gumbel-max couplings are counterfactually stable. We might therefore hope that they are also maximal couplings, i.e. that they assign as much probability as possible to the counterfactual and observed outcomes being the same. Unfortunately, it turns out this is not the case.

**Proposition 1.** *The probability that $x = y = i$ in a Gumbel-max coupling is $\frac{1}{1+\sum_{j\neq i}\max(\frac{p(j)}{p(i)},\frac{q(j)}{q(i)})}$.*

The full proof is in Appendix C. The main idea is to express the event $x = y = i$ as a conjunction of inequalities defining the argmax, then simplifying using properties of Gumbel distributions.

**Corollary 1.** *Gumbel-max couplings are not maximal couplings. In particular, they are suboptimal under the $\mathbb{1}_{x\neq y}$ metric iff there is an $i$ such that*

$$\max\left(\sum_{j\neq i}\frac{p(j)}{p(i)}, \sum_{j\neq i}\frac{q(j)}{q(i)}\right) < \sum_{j\neq i}\max\left(\frac{p(j)}{p(i)},\frac{q(j)}{q(i)}\right). \tag{9}$$

The proof appears in Appendix D. It follows from Prop. 3 and the fact that the probability of $x = y = i$ in a maximal coupling is $\min(p(i),q(i))$. On the positive side, it is straightforward to show that Gumbel-max couplings are optimal when $p = q$ and when there are only two possible outcomes (in which case they also satisfy the monotonicity assumption of Pearl [1999]). We also show that Gumbel-max couplings are within a constant-factor of optimal as maximal couplings.

**Corollary 2.** *If the Gumbel-max coupling assigns probability $\alpha$ to the event that $x = y$, then the probability that $x = y$ under the maximal coupling is at most $2\alpha$.*

The proof appears in Appendix E. It comes from bounding the ratio of the LHS to the RHS in Eq. 9.

## 4.2 Impossibility of Implicit Maximal Couplings

Since Gumbel-max does not always induce the maximal coupling, we might wonder if some other implicit coupling mechanism could. Here we show that it is impossible. In particular, we show that no fixed implicit coupling is maximal for every pair of distributions over the set $\{1, 2, 3\}$. Thus, there will always be some pair of distributions for which an implicit coupling is non-maximal. A proof of the Proposition appears in Appendix F.

**Proposition 2.** *There is no algorithm $f_\theta$ that takes logits $l$ and a sample $\mathbf{u}$ of noise, and deterministically transforms $\mathbf{u}$ into a sample from $\exp l$, such that when given any two distributions $p$ and $q$ and using shared noise $u$, the joint distribution of samples is always a maximal coupling of $p$ and $q$.*

# 5 Methods for Learning Implicit Couplings

Here we develop methods for learning implicit couplings, the problem defined in Sec. 3. We use the term *gadget* to refer to a learnable, continuously parameterized family of $f_\theta$. We present two gadgets in this section. Gadget 1 does not fully satisfy the requirements laid out in Sec. 3, but it is simpler and introduces some key ideas, so we present it as a warm-up. Gadget 2 fully satisfies the requirements.

## 5.1 Gadget 1

The main idea in Gadget 1 is to learn a mapping $\pi_\theta : \mathbb{R}^K \to \mathbb{R}^{K \times K}$ from categorical distribution $p \in \mathbb{R}^K$ to an auxiliary joint distribution $\pi_\theta(x, z|p)$, represented as a matrix $\pi_\theta(\cdot, \cdot|p) \in \mathbb{R}^{K \times K}$. The architecture of the mapping is constrained so that marginalizing out the auxiliary variable $z$ yields a distribution consistent with the given logits, i.e., $\sum_z \pi_\theta(x, z|p) = p(x)$. We then generate $K^2$ independent $\gamma_{x,z} \sim \text{Gumbel}(0)$ samples and perform Gumbel-max on the auxiliary joint. We only care about the sample of $x$, so one way of doing this is to first maximize out the auxiliary dimension to get $\gamma(p) = \max_z \{\gamma_{x,z} + \log \pi_\theta(x, z|p)\}$ and then return $\hat{x} = \text{argmax}\{\gamma(p)\}$. Here $\hat{x}$ is distributed according to $p$ because this is performing Gumbel-max on a joint distribution with correct marginals and then marginalizing out the auxiliary variable.

To create a coupling, we run this process separately for $p$ and $q$ but with shared realizations of the $K^2$ Gumbels. However, the place where Gadget 1 does not fully satisfy the requirements from Sec. 3 is that we transpose the Gumbels for one of the samples. That is, we sample a coupling as

$$[\gamma_1(p)]_x = \max_z \{\gamma_{x,z} + \log \pi_\theta(x, z|p)\} \qquad [\gamma_2(q)]_y = \max_z \{(\gamma^T)_{y,z} + \log \pi_\theta(y, z|q)\} \qquad (10)$$

$$\hat{x} = \text{argmax}\{\gamma_1(p)\} \qquad\qquad \hat{y} = \text{argmax}\{\gamma_2(q)\}. \qquad (11)$$

Gadget 1 can still be used to create a coupling, but it is more analogous to antithetical sampling, where the noise source is used differently based on which sample is being drawn. Note that like in antithetical sampling, both processes draw samples from the correct distribution, since transposing a matrix of independent Gumbels does not change the distribution. We describe how to draw counterfactual samples from Gadget 1 in Appendix G.

We note that if $\theta$ is chosen such that $\pi_\theta(x = k, z = k|p) = p(x = k)$ and $\pi_\theta(x, z|p) = 0$ for $z \neq x$, the Gadget 1 SCM becomes identical to the Gumbel-max SCM. Thus, Gadget 1 is a generalization of the Gumbel-max causal mechanism.

## 5.2 Gadget 2

Gadget 1 is not an implicit coupling as defined in Sec. 3, because it requires Gumbels to be transposed when sampling $p$ versus $q$. In this section, we present a gadget that is a proper implicit coupling.

Gadget 2 again invokes an auxiliary variable and parameterizes a learned joint distribution, but the auxiliary variable $z$ is no longer required to share the same sample space as $x$. Further, instead of performing Gumbel-max on the learned joint directly, we start by drawing a single $z$ independently of $p$. The gadget is defined as follows:

$$\gamma_i^{(z)} \sim \text{Gumbel}(0) \ \text{ for } i = 1, \ldots, |Z| \qquad \gamma_i^{(x)} \sim \text{Gumbel}(0) \ \text{ for } i = 1, \ldots, |X| \qquad (12)$$

$$\hat{z} = \text{argmax}_z(\log \pi_\theta(z) + \gamma^{(z)}) \qquad\qquad \hat{x} = \text{argmax}_x(\log \pi_\theta(x \mid \hat{z}, p) + \gamma^{(x)}). \qquad (13)$$

To sample a coupling, we re-use all the $\gamma$'s and run the same process with $q$ instead of $p$. This means that we additionally get $\hat{y} = \mathrm{argmax}_y(\log \pi_\theta(y \mid \hat{z}, q)) + \gamma^{(x)})$. Intuitively, we can think of $\hat{z}$ as a latent cluster identity and each cluster being associated with a different learned mapping $\pi_\theta(x \mid \hat{z}, p)$. The learning can choose how to assign clusters so that a Gumbel-max coupling of $\pi_\theta(x \mid \hat{z}, p)$ and $\pi_\theta(y \mid \hat{z}, q)$ produces joint outcomes that are favorable under $g$.

**Architecture for $\pi_\theta(x|z,p)$.** Not all choices of $\pi_\theta(x|z,p)$ lead to correct samples. For correctness, we need to enforce the analogous constraint as in Gadget 1, which is that when we integrate out the auxiliary variable, we get samples from the $p$ distribution provided as an input; i.e., $\sum_z \pi_\theta(z)\pi_\theta(x|z,p) = p(x)$ for all $p$. Here we describe how to build an architecture for $\pi_\theta(x|z,p)$ that is guaranteed to satisfy the constraint.

First, we use a neural function approximator $h_\theta : \mathbb{R}^K \to \mathbb{R}_+^{Z \times K}$ that maps logits $l = \log p$ to a nonnegative matrix $A_0 = h_\theta(l)$ of probabilities for each pair $(z, x)$. Next, we iteratively normalize the columns to have marginals $p(x)$ and the rows of $A$ to have marginals $\pi_\theta(z)$ for $T$ steps (a modified version of the algorithm proposed by Sinkhorn and Knopp [1967]). The last iterate $A = A_T$ always satisfies $\sum_x A_{x,z} = \pi_\theta(z)$ but may only approximately satisfy the constraint that $\sum_z \pi_\theta(z)\pi_\theta(x|z,p) = p(x)$. To deal with this, we treat $A$ as a proposal and apply a final accept-reject correction, falling back to an independent draw from $p$ if the $z$-dependent proposal is rejected. The marginals of this process give our expression for $\pi_\theta(x|z,p)$:

$$c_x = \frac{p(x)}{\sum_z A_{x,z}}, \qquad d_z = \sum_x \frac{A_{x,z}}{\pi_\theta(z)} c_x, \qquad \pi_\theta(x|z,p) = \frac{c_x}{c_*} \cdot \frac{A_{x,z}}{\pi_\theta(z)} + \left(1 - \frac{d_z}{c_*}\right) p(x) \qquad (14)$$

where $c_* = \max_x c_x$. Encoding this expression in the architecture of $\pi_\theta(x|z,p)$ ensures that $\sum_z \pi_\theta(z)\pi_\theta(x|z,p) = p(x)$, and thus all choices of $\theta$ yield a valid reparameterization trick for all $p$. See Appendix H for a proof. While we could parameterize and learn $\pi_\theta(z)$, we have thus far fixed it to the uniform distribution.

We note that if we let $|Z| = 1$ (i.e. we assign all outcomes to one cluster), $\hat{z}$ becomes deterministic, and thus $\pi_\theta(x \mid \hat{z}, p) = \pi_\theta(x \mid p) = p(x)$. In this case, we recover a Gumbel-max coupling of $p(x)$ and $q(y)$, showing that Gadget 2 also generalizes the Gumbel-max SCM.

**Sampling from counterfactual distributions.** Given a particular outcome $x \sim p$, we can sample a counterfactual $y$ under some intervention $q$ by first computing the posterior $\pi_\theta(z|x, l^{(1)}) \propto \pi_\theta(z)\pi_\theta(x|z, l^{(1)})$ and sampling an auxiliary variable $z$ that is consistent with the observation. Given $z$, we obtain a Gumbel-max coupling between $\pi_\theta(x|z,p)$ and $\pi_\theta(y|z,q)$, so the top-down algorithm from Oberst and Sontag [2019] can be used to sample Gumbels and a counterfactual outcome $y$.

## 5.3 Learning Gadgets

Recall from Sec. 3 that our goal is to learn $\theta$ so that the gadgets above produce favorable implicit couplings when measured against dataset $\mathcal{D}$ and cost function $g$. In both gadgets, the constraint in Eq. 8 is automatically satisfied by the architecture. Thus, we need only concern ourselves with Eq. 7, which is a minimization problem over $\mathcal{L}$ with the following form:

$$\mathcal{L}(\theta) = \mathbb{E}_{(p,q)\sim\mathcal{D}}\mathbb{E}_\gamma [g(f_\theta(\gamma, p), f_\theta(\gamma, q))] \qquad (15)$$

We would like to use a reparameterization gradient where we sample $(p, q)$ and $\gamma$, and then differentiate the inner term with respect to $\theta$. However, the loss is not a smooth function of $\theta$ given a realization of $\gamma$ due to the argmax operations in Eqs. 11, 13. Thus, our learning strategy is to relax these argmax operations into softmaxes, as in Jang et al. [2017], Maddison et al. [2017]. This yields a smoothed $\tilde{f}_\theta \in \Delta^{K-1}$ and a smoothed softmax surrogate loss:

$$\tilde{\mathcal{L}}(\theta) = \mathbb{E}_{(p,q)\sim\mathcal{D}}\mathbb{E}_\gamma \left[ \sum_{x,y} [\tilde{f}_\theta(\gamma, p)]_x \cdot [\tilde{f}_\theta(\gamma, q)]_y \cdot g(x, y) \right]. \qquad (16)$$

This is differentiable, and we can optimize it using gradient based methods and standard techniques (either explicitly summing over all $x, y$ or taking a sample-based REINFORCE gradient).

# 6 Related Work

The variational approach for coupling relates the maximal coupling to the total variation distance $\|p - q\|_{TV} \triangleq \max_{A \subset \{1,...,K\}} |p(A) - q(A)|$ since $\|p - q\|_{TV} = \inf_{\pi \in C(p,q)} \mathbb{P}_\pi[X \neq Y]$. The Wasserstein distance $W(p, q; d)$ in Eq. 5, is a generalization of the variational principle. The Wassestein distance can be extended to the optimal transport setting, when $d$ is any function, which has been used extensively in machine learning, see [Frogner et al., 2015, Arjovsky et al., 2017, Alvarez-Melis et al., 2018, Benamou et al., 2015, Blondel et al., 2018, Courty et al., 2016, 2017, Cuturi, 2013, Aude et al., 2016, Kusner et al., 2015, Luise et al., 2018, Peyré et al., 2019].

The maximal coupling of Bernoulli random variables enforces monotonicity: it is attained by sampling $u \sim Uniform([0, 1])$ and setting $X = 1[u \leq p]$ and $Y = 1[u \leq q]$ [Fréchet, 1951]. More generally, Strassen's theorem asserts that any two random variables satisfy $F_X(t) \leq F_Y(t)$ if and only if they are monotone, i.e., there is a coupling $\pi$ for which $\mathbb{P}_\pi[Y \leq X] = 1$. The monotone coupling can be realized by using the same uniform random variable U, setting $X = F_X^{-1}(U)$ and $Y = F_Y^{-1}(U)$ [Lindvall et al., 1999]. A monotone coupling $\pi \in C(p, q)$ of two marginal probabilities $p$ and $q$, maximizes the covariance of $(X, Y)$ and consequently minimize the variance of $X - Y$, since $Var_\pi[X - Y] = Var_p[X] + Var_q[\hat{Y}] - 2Cov_\pi(\hat{X}, \hat{Y})$. This is equivalent to maximizing the correlation between $X$ and $Y$. Minimizing the variance of $X - Y$ helps to stabilize their estimation in machine learning applications, e.g., Grathwohl et al. [2017].

Cambanis et al. [1976] give conditions when the coupling $(F_X^{-1}(U), F_Y^{-1}(U))$ is optimal. Let $d(x, y)$ be *supermodular*, i.e., $d(x_1, y_1) + d(x_2, y_2) \geq d(x_1, y_2) + d(x_2, y_1)$ if $x_1 \leq x_2$ and $y_1 \leq y_2$. Then $\sup_{\pi \in C(p,q)} \mathbb{E}_{x,y \sim \pi} d(x, y) = \mathbb{E}_{U \sim \text{Uniform}(0,1)} d(F_X^{-1}(U), F_Y^{-1}(U))$. Specifically, when $d(x, y) = h_1(x)h_2(y)$ and $h_i(F_X^{-1}(u))$ are non-increasing or non-decreasing functions of $u$, the LHS is the coupling with maximum covariance and the RHS is the coupling achieved by common random numbers paired with a CDF inversion mechanism. See also Glasserman and Yao [1992].

Monotonicity assumptions have also been used in the causality literature to enable identification of a unique level 3 SCM from interventional data, for both binary [Pearl, 1999] and categorical [Lu et al., 2020] random variables. We note that the implicit coupling induced by these assumptions also corresponds to the inverse-CDF coupling $(F_X^{-1}(U), F_Y^{-1}(U))$.

Li and Anantharam [2019] consider joint couplings of a collection of distributions that are within a constant factor of optimal for all pairs. Their approach uses Poisson processes, which are closely related to Gumbel-max [Maddison, 2016], and they introduce auxiliary latent variables to adapt the coupling to particular cost functions, similar to our gadgets. Unlike this work, Li and Anantharam do not impose a distribution over the collection of distributions or interpret the coupling as a SCM, and they focus on directly constructing couplings instead of learning them.

# 7 Experiments

We evaluate our approach in two settings.[3] First, we build understanding by exploring performance on several datasets of fixed and random logits $\in R^K$. Next, we learn low-variance counterfactual treatment effects in the sepsis management MDP from Oberst and Sontag [2019].

## 7.1 Optimizing for maximality

Section 4.2 shows that no implicit coupling is maximal for every pair of $p$ and $q$. Here we show that we can learn near-maximal couplings if we limit attention to narrower distribution $\mathcal{D}$ of interest. We compare our proposed learning method against fixed Gumbel-max couplings and a maximal coupling. We hold fixed a single pair of test distributions $p^{test}$ and $q^{test}$ and then vary distributions that are trained on. Specifically, we train Gadget 2 on pairs $(p^{test} + \rho \cdot \eta_p, q^{test} + \rho \cdot \eta_q)$ where $\eta_p$ and $\eta_q$ are vectors of unit variance Gaussian noise, and $\rho$ is a noise scale. When $\rho$ is small, it is possible to learn an implicit coupling that is specific to the region of distributions around $(p^{test}, q^{test})$ and we can achieve implicit couplings that are nearly maximal. When $\rho$ is large, we are asking $f_\theta$ to couple together all pairs of distributions, and we expect to run into the impossibility results in Sec. 4.2.

---

[3]An implementation of our approach and instructions for reproducing our experiments is available at `https://github.com/google-research/google-research/tree/master/gumbel_max_causal_gadgets`.

Table 1: Comparison of gadgets in controlled setting, shown with standard error.

| Reward $h$: | Fixed linear ($h(x) = x$) | | Random monotonic | Non-monotonic |
|---|---|---|---|---|
| $p$ and $q$: | Independent | Mirrored | Fixed inc/dec | Fixed inc/dec |
| Independent | $16.51 \pm 0.01$ | $21.34 \pm 0.10$ | $2.73 \pm 0.39$ | $0.83 \pm 0.10$ |
| Gumbel-max | $14.02 \pm 0.01$ | $18.84 \pm 0.09$ | $2.46 \pm 0.35$ | $0.50 \pm 0.06$ |
| $CDF^{-1}$ | $8.14 \pm 0.01$ | $12.59 \pm 0.09$ | $0.60 \pm 0.10$ | $0.91 \pm 0.12$ |
| Optimal (LP) | $8.14 \pm 0.01$ | $12.59 \pm 0.09$ | $0.60 \pm 0.10$ | $0.11 \pm 0.02$ |
| Gadget 1 | $14.05 \pm 0.01$ | $16.67 \pm 0.45$ | $1.42 \pm 0.22$ | $0.26 \pm 0.03$ |
| Gadget 2 | $8.76 \pm 0.03$ | $13.47 \pm 0.09$ | $2.00 \pm 0.30$ | $0.21 \pm 0.03$ |

Results appear in Fig. 1 (a), with additional visualizations in Appendix I. Indeed, when noise scales are small, our gadget learns a near-maximal coupling. When the noise scale becomes large, the quality declines. Interestingly, when the noise scale is orders of magnitude larger than the signal, our gadget never becomes significantly worse than the quality of Gumbel-max couplings.

## 7.2 Minimizing variance over random logits

In this experiment, we consider the ability of our learned couplings to reduce variance in a controlled setting. Specifically, we define a scalar "reward" $h(x)$ for each outcome $x$, and attempt to minimize the variance of $\mathbb{E}_{x \sim p}[h(x)] - \mathbb{E}_{y \sim q}[h(y)] = \mathbb{E}_{x,y \sim \pi}[h(x) - h(y)]$. We explore both randomness in $p$ and $q$ and randomness in the reward $h$, and show that our gadgets achieve a lower variance than both the inverse CDF and Gumbel-max methods under non-monotonic reward functions.

In the first part, we fixed the reward to the identity function and randomized $p$ and $q$ in two ways: (1) independently randomly drawn $p$ and $q$, (2) $p$ drawn randomly and $q$ set to have the same probabilities as $p$ but assigned in a mirrored order. This tests the abilities of our gadgets to learn to couple arbitrary distributions and to uncover relationships between $p$ and $q$.

In the second part, we test our two gadgets under varying monotonic and non-monotonic reward functions $h$. We fix $p$ to be linearly increasing and $q$ to be linearly decreasing (the reverse of $p$), to examine how the gadgets perform when $p$ is very different from $q$. The monotone function is constructed by taking the cumulative sum of a K-length vector of $\text{Uniform}(0, 1)$. The non-monotone function is constructed by sampling K-length vector from a Gaussian distribution followed by the function $R(i) = \sin(30 * i)$. At each trial, the gadgets are trained from scratch under a new realization of rewards. For comparison, we also solve for the optimal coupling using linear programming [Bertsimas and Tsitsiklis, 1997] (which may not be achievable by any implicit coupling).

Table 1 shows the results of our experiments. We find that Gadget 2 shows strong performance across all distributions of $p$ and $q$, outperforming the Gumbel-max and independent sampling, and approaching the results of the optimal coupling under non-monotone $h$. Gadget 1 outperforms Gumbel-max in the mirrored setting, and also outperforms Gadget 2 in the fixed increasing $p$ / decreasing $q$ setting under monotonic $h$.

## 7.3 MDP counterfactual treatment effects

In this experiment, we use a synthetic environment of sepsis management to minimize the variance of counterfactual treatment effects on an SCM for MDP. Following Oberst and Sontag [2019], we used the simulator only for obtaining the initial observed trajectories and we do not assume access to the simulator to get counterfactual probabilities. The simulator includes four vital signs (blood pressure, oxygen concentration, and glucose levels) with discrete states (low, normal, high), as well as three treatment options (antibiotics, vasopressors, and mechanical ventilation). Our goal is to couple $p(s'|s, a_{doctor})$ and $p(s'|s, a_{intervention})$, i.e., the transition distributions induced by two policies: a behavior policy, which mimics the physician policy, and a target policy, which is the RL policy. Our counterfactual question is what would have happened to the patient if the RL policy's action had been applied instead of the doctor's.

Using a trained behavior policy, we draw 20,000 patient trajectories from the simulator with a maximum of 20 time steps, where the states are in a space of 6 discrete variables, each with different

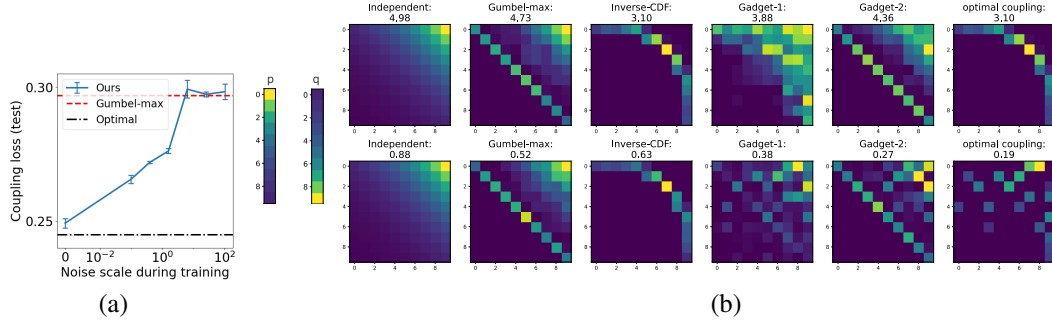

Figure 1: (a) When the training distribution is focused, we learn a near-maximal coupling. As the distribution becomes more diffuse, the learned coupling reverts to Gumbel-max. (b) Couplings of each method for the increasing $p$ / decreasing $q$ settings along with the counterfactual effect variance for this specific reward realization. First row: monotone reward. Second row: non-monotone reward.

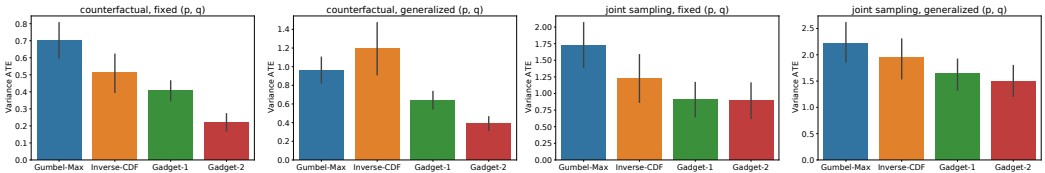

Figure 2: Variance of the treatment effect on the sepsis management simulator.

number of categories (146 states in total). Based on them, we learn the parameters of an MDP, and train the target policy over this MDP. Unlike Oberst and Sontag [2019], we focused on coupling single time steps. We set the total reward for each state by summing its discrete variables rewards, which were sampled from a Gaussian distribution. Among all the trajectories and time steps we filtered out all pairs that have less than 4 non-zero probabilities. We conducted the experiment in two settings: joint sampling, where the noise is shared between the two samples, and counterfactual sampling, where we infer the noise $u^{(env)}$ based on the observation $(s, a_{doctor}, s')$, then sample $(s'|s, a_{intervention})$. In each setting, we tested our gadgets when $(p, q)$ are fixed and when $(p, q)$ are perturbed by a Gaussian sample (generalized). At testing, we compute the treatment effect variance over 2000 samples and average the result across 10 different random seeds. With each trial of the experiment we fixed a pair of $(p, q)$ and set a new random realization of reward. We repeated that process for 6 pairs of $(p, q)$ and 5 reward realizations, a total of 30 trials for each setting.

The means and the standard errors of the experiments are shown in Figure 2. Both our gadgets outperformed the fixed sampling mechanisms, and Gadget 2 did so by a significant margin under the counterfactual settings. Details on the implementation of all the experiments are in Appendix I.

# 8 Discussion

We have presented methods for learning a level 3 causal mechanism that is consistent with observed level 2 information. Our framework provides significant flexibility to quantitatively define what makes a causal mechanism desirable, and then uses learning to find the best mechanism. Since any such choice cannot be confirmed or rejected by running an experiment, one might argue that choosing an SCM this way is unprincipled. However, principles such as counterfactual stability can still be encoded into our framework using the loss $g$. We thus see our gadgets as powerful tools which give modelers both the freedom and also the responsibility to select appropriate criteria for causal inference tasks, instead of being restricted to assumptions that cannot be tuned for a specific use case.

One limitation is that we have only considered single-step causal processes, but we believe the framework can be extended to multi-step MDPs. In future work, we would also like to explore other settings of $\mathcal{D}$ and $g$ and investigate their qualitiative properties, such as how intuitive the resulting counterfactuals are to humans.

## Acknowledgements

We would like to thank the anonymous reviewers of our submission, whose excellent suggestions and requests for clarification were very helpful for improving the paper. We would also like to thank Michael Oberst and David Sontag for providing their implementation of the sepsis simulator and off-policy evaluation logic, which we used for our experiments in Section 7.3. This research was supported by Grant No. 2029378 from the United States-Israel Binational Science Foundation (BSF).

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
