# Supplementary Material: Learning Generalized Gumbel-max Causal Mechanisms

## A  Diagrams of Gadget 1 and Gadget 2

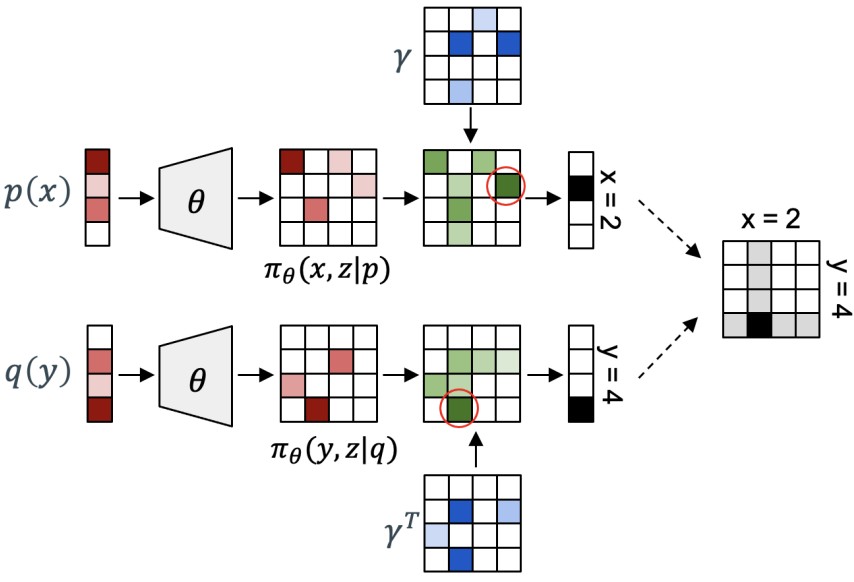

Figure 3: Diagram of Gadget 1 applied to an observed distribution $p(x)$ and counterfactual distribution $q(x)$, resulting in a coupled pair of observations. Note that $\gamma$ is transposed.

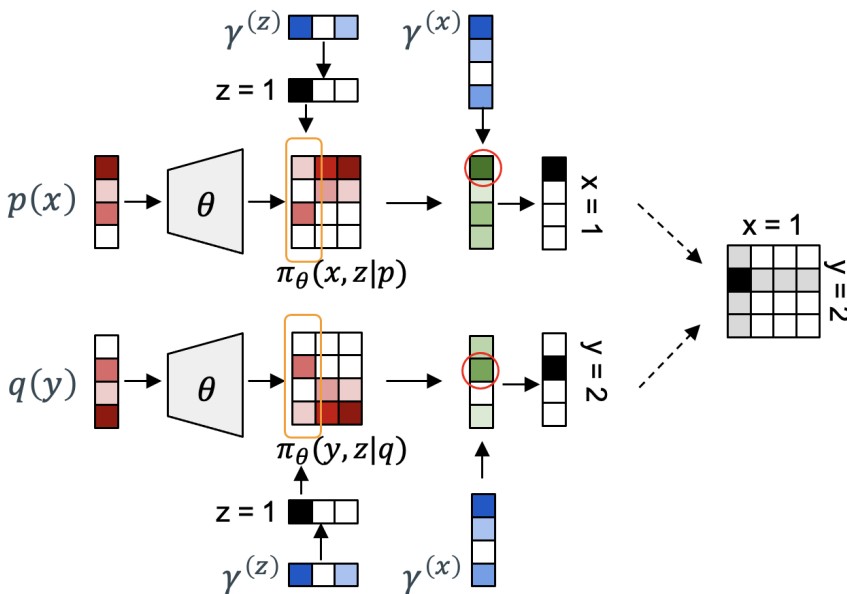

Figure 4: Diagram of Gadget 2 applied to an observed distribution $p(x)$ and counterfactual distribution $q(x)$, resulting in a coupled pair of observations. Note that $\gamma^{(z)}$, $z$, and $\gamma^{(x)}$ are shared.

# B  Proof of expected difference of costs equation for Gumbel-max SCM

In this section we show that

$$\mathbb{E}_{x,y\sim\pi}[h_1(x)-h_2(y)] = \sum_y q(y)\mathbb{E}_{\gamma\sim(G_2|y)}\Big[h_1(\mathrm{argmax}_k[l_k^{(1)}-l_k^{(2)}+\gamma_k]) - h_2(\mathrm{argmax}_k[\gamma_k])\Big].$$

Here $G_2|y$ is the Gumbel distribution with location $\log l_k^{(2)}$ conditioned on the event that $y$ is the maximal argument. We start by observing that

$$\mathbb{E}_{x,y\sim\pi}[h_1(x)-h_2(y)] = \mathbb{E}_{x\sim p}[h_1(x)] - \mathbb{E}_{y\sim q}[h_2(y)]$$

Considering each term separately, we find

$$
\begin{aligned}
\mathbb{E}_{y\sim p}[h_1(y)] &= \mathbb{E}_{\gamma\sim G_1}\Big[h_1(\mathrm{argmax}_k[\gamma_k])\Big] = \mathbb{E}_{\gamma\sim G_0}\Big[h_1(\mathrm{argmax}_k\Big[\gamma_k + l_k^{(1)}\Big])\Big]\\
&= \mathbb{E}_{\gamma\sim G_0}\Big[h_1(\mathrm{argmax}_k\Big[\gamma_k + l_k^{(2)} + (l_k^{(1)} - l_k^{(2)})\Big])\Big]\\
&= \mathbb{E}_{\gamma\sim G_2}\Big[h_1(\mathrm{argmax}_k\Big[\gamma_k + (l_k^{(1)} - l_k^{(2)})\Big])\Big]\\
&= \mathbb{E}_{\gamma\sim G_2}\Big[\sum_y 1[y = \mathrm{argmax}_k[\gamma_k]]h_1(\mathrm{argmax}_k\Big[\gamma_k + (l_k^{(1)} - l_k^{(2)})\Big])\Big]\\
&= \sum_y q(y)\mathbb{E}_{\gamma\sim(G_2|y)}\Big[\tfrac{1[y=\mathrm{argmax}_k[\gamma_k]]}{q(y)}h_1(\mathrm{argmax}_k\Big[\gamma_k + (l_k^{(1)} - l_k^{(2)})\Big])\Big]\\
&= \sum_y q(y)\mathbb{E}_{\gamma\sim(G_2|y)}\Big[h_1(\mathrm{argmax}_k\Big[\gamma_k + (l_k^{(1)} - l_k^{(2)})\Big])\Big]\\
\mathbb{E}_{y\sim q}[h_2(y)] &= \mathbb{E}_{\gamma\sim G_2}\Big[h_1(\mathrm{argmax}_k[\gamma_k])\Big]\\
&= \mathbb{E}_{\gamma\sim G_2}\Big[\sum_y 1[y = \mathrm{argmax}_k[\gamma_k]]h_1(\mathrm{argmax}_k[\gamma_k])\Big]\\
&= \sum_y q(y)\mathbb{E}_{\gamma\sim(G_2|y)}\Big[h_1(\mathrm{argmax}_k[\gamma_k])\Big]
\end{aligned}
$$

Combining these terms yields the desired equation.

# C  Probability of $x = y$ in Gumbel-max Coupling

**Proposition 3.** *The probability that $x = y = i$ in a Gumbel-max coupling is $\frac{1}{1+\sum_{j\neq i}\max(\frac{p(j)}{p(i)},\frac{q(j)}{q(i)})}$.*

*Proof.* The event that $x = y = i$ is equivalent to

$$\log p(i) + \gamma(i) > \log p(j) + \gamma(j) \text{ for all } j \neq i \text{ and} \qquad (17)$$
$$\log q(i) + \gamma(i) > \log q(j) + \gamma(j) \text{ for all } j \neq i. \qquad (18)$$

Or equivalently,

$$\gamma(i) > \log p(j) - \log p(i) + \gamma(j) \text{ for all } j \neq i, \text{ and} \qquad (19)$$
$$\gamma(i) > \log q(j) - \log q(i) + \gamma(j) \text{ for all } j \neq i. \qquad (20)$$

Only one inequality per $j$ can be active, so we can combine to get the equivalent event

$$\gamma(i) > \log\max(\frac{p(j)}{p(i)}, \frac{q(j)}{q(i)}) + \gamma(j) \text{ for all } j \neq i. \qquad (21)$$

By the Gumbel-max trick, the probability of this event is a softmax $\frac{1}{1+\sum_{j\neq i}\max(\frac{p(j)}{p(i)},\frac{q(j)}{q(i)})}$. $\qquad\square$

# D  Suboptimality of Gumbel-max Couplings

**Corollary 3.** *Gumbel-max couplings are not maximal couplings. In particular, they are suboptimal when*

$$\max(\sum_{j\neq i}\frac{p(j)}{p(i)}, \sum_{j\neq i}\frac{q(j)}{q(i)}) < \sum_{j\neq i}\max(\frac{p(j)}{p(i)}, \frac{q(j)}{q(i)}). \tag{22}$$

*Proof.* The probability that $x = y = i$ in a maximal coupling is $\min(p(i), q(i))$. Note that we can rewrite

$$p(i) = \frac{p(i)}{\sum_j p(j)} = \frac{1}{1 + \sum_{j\neq i}\frac{p(j)}{p(i)}} \tag{23}$$

$$q(i) = \frac{q(i)}{\sum_j q(j)} = \frac{1}{1 + \sum_{j\neq i}\frac{q(j)}{q(i)}}, \qquad \text{and} \tag{24}$$

$$\min(p(i), q(i)) = \frac{1}{1 + \max(\sum_{j\neq i}\frac{p(j)}{p(i)}, \sum_{j\neq i}\frac{q(j)}{q(i)})}. \tag{25}$$

Comparing this expression to Proposition 1, this means Gumbel-max couplings are suboptimal when

$$\max(\sum_{j\neq i}\frac{p(j)}{p(i)}, \sum_{j\neq i}\frac{q(j)}{q(i)}) < \sum_{j\neq i}\max(\frac{p(j)}{p(i)}, \frac{q(j)}{q(i)}). \tag{26}$$

$\square$

# E  Constant Factor Approximation of Gumbel-max Couplings

**Corollary 4.** *If the Gumbel-max coupling assigns probability $\alpha$ to the event that $x = y$, then the probability that $x = y$ under the maximal coupling is at most $2\alpha$.*

*Proof.* First expand out the above equations and replace 1 with $\frac{p(i)}{p(i)}$ or $\frac{q(i)}{q(i)}$:

$$\frac{\min(p(i), q(i))}{\pi_{gm}(x = i, y = i)} = \frac{1 + \sum_{j\neq i}\max(\frac{p(j)}{p(i)}, \frac{q(j)}{q(i)})}{1 + \max(\sum_{j\neq i}\frac{p(j)}{p(i)}, \sum_{j\neq i}\frac{q(j)}{q(i)})} = \frac{\sum_j \max(\frac{p(j)}{p(i)}, \frac{q(j)}{q(i)})}{\max(\sum_j \frac{p(j)}{p(i)}, \sum_j \frac{q(j)}{q(i)})} \tag{27}$$

Then we can prove the bound, leveraging the fact that all $p$ and $q$ are positive:

$$\frac{\sum_j \max(\frac{p(j)}{p(i)}, \frac{q(j)}{q(i)})}{\max(\sum_j \frac{p(j)}{p(i)}, \sum_j \frac{q(j)}{q(i)})} \leq \frac{\sum_j \frac{p(j)}{p(i)} + \sum_j \frac{q(j)}{q(i)}}{\max(\sum_j \frac{p(j)}{p(i)}, \sum_j \frac{q(j)}{q(i)})} \leq \frac{2\max(\sum_j \frac{p(j)}{p(i)}, \sum_j \frac{q(j)}{q(i)})}{\max(\sum_j \frac{p(j)}{p(i)}, \sum_j \frac{q(j)}{q(i)})} = 2. \tag{28}$$

Since this is true for each $i$, it implies that the total probability of $x = y$ in the maximal coupling is at most twice that of the Gumbel-max coupling. $\square$

# F  Proofs of Impossibility of Maximal Couplings

**Proposition 4.** *Let $\Omega$ be a probability space with measure $\mu$, and $F_p : \Omega \to \{1, 2, 3\}$ be a family of functions, indexed by a marginal distribution $p \in \Delta^3$ over three choices, that maps events $\boldsymbol{u} \in \Omega$ into three outcomes. Let $F_p^{-1}(i) = \{\boldsymbol{u} \in \Omega \ : \ F_p(\boldsymbol{u}) = i\}$. Then the following cannot both be true:*

1. *F assigns correct marginals, i.e.*

$$\mu(F_p^{-1}(1)) = p_1, \qquad \mu(F_p^{-1}(2)) = p_2, \qquad \mu(F_p^{-1}(3)) = p_3. \tag{29}$$

2. *$\mu(F_p^{-1}(i) \cap F_q^{-1}(i)) = \min(\mu(F_p^{-1}(i)), \mu(F_q^{-1}(i)))$ for all $p, q, i$. Note that this implies $F_q^{-1}(i) = F_p^{-1}(i)$ whenever $q_i = p_i$, except possibly on a set of measure zero.*

*Proof.* Suppose $F$ satisfies both. Let

$$A_{12} = F^{-1}_{[\frac{1}{2},\frac{1}{2},0]}(1), \qquad B_{12} = F^{-1}_{[\frac{1}{2},\frac{1}{2},0]}(2), \qquad C_{13} = F^{-1}_{[\frac{1}{2},0,\frac{1}{2}]}(3),$$

$$A_{13} = F^{-1}_{[\frac{1}{2},0,\frac{1}{2}]}(1), \qquad B_{23} = F^{-1}_{[0,\frac{1}{2},\frac{1}{2}]}(2), \qquad C_{23} = F^{-1}_{[0,\frac{1}{2},\frac{1}{2}]}(3),$$

From 1, all of these sets have measure $\frac{1}{2}$, and from 2, we know $A_{12} = A_{13}, B_{12} = B_{23}, C_{13} = C_{23}$. Also, $A_{12} \sqcup B_{12} = A_{13} \sqcup C_{13} = \Omega$, so we must have $B_{12} = C_{13}$. But then $B_{12} = B_{12} \sqcup C_{13} = B_{23} \sqcup C_{23} = \Omega$ and so $\mu(\Omega) = \frac{1}{2}$, which is impossible since $\mu(\Omega) = 1$. □

**Proposition 5.** *There is no reparameterization algorithm $f_\theta$ that takes logits $l$ and a sample $\mathbf{u}$ of noise, and deterministically transforms $\mathbf{u}$ into a sample from $\exp l$, such that when given any two distributions $p$ and $q$ and using shared noise $u$, the joint distribution of samples is always a maximal coupling of $p$ and $q$.*

*Proof.* Suppose such an algorithm existed. Choose $\Omega = \mathbb{R}^D$ with the appropriate measure $\mu$, and let $F_p(\mathbf{u}) = f_\theta(\mathbf{u}, \log p)$. By assumption, the algorithm must satisfy 1, so there must be some $p, q$ and $i$ that do not satisfy 2. But then $f_\theta$ does not produce a maximal coupling between $p$ and $q$: in particular we must have $\pi_\theta(i, i) \neq \min(p(i), q(i))$. □

# G   Sampling from counterfactual distributions in Gadget 1

To use the gadget in a counterfactual setting, we need to condition on an outcome $x^{(obs)}$ and then draw a counterfactual sample of $y$. As in the counterfactual sampling algorithm for Gumbel-max couplings [Oberst and Sontag, 2019], we rely on top-down Gumbel sampling [Maddison et al., 2014], but here we require a slightly more elaborate two-step construction. Due to properties of Gumbels, we know that $[\gamma_1(p)]_x \sim \text{Gumbel}(\log \sum_z \pi_\theta(x, z|p))$. We can then sample the $\gamma_1(p)$ conditioned on $x^{(obs)}$ being the argmax using top-down sampling. Next, using the independence of the argmax and max, we can independently sample $z_x^* = \text{argmax}_z \gamma_{x,z} + \log \pi_\theta(x, z|p)$ for each $x$ by sampling from the distribution $\pi_\theta(x|p) = \sum_z \pi_\theta(x, z|p)$. Finally, we combine $[\gamma_1(p)]_x$ (the maximum over $z$) and $z_x^*$ (the argmax) using top-down sampling to obtain the values of $\gamma_{x,z}$. The result is a posterior sample of the full $K \times K$ matrix $\gamma_{x,y}$ of exogenous Gumbel noise that is consistent with the behavior policy. To get a counterfactual sample, transpose the $\gamma_{x,z}$'s and run the $q$ gadget forward to sample $y$.

# H   Proof that the accept-reject correction in Gadget 2 produces the desired marginals

Here we show that the correction step described in Section 5.2 produces the correct marginals. We start by rewriting the correction in terms of possibly unnormalized log probabilities $l$, for which $p(x) \propto \exp l_x$. Note that we omit the normalization constant in the definition of $c_x$ (writing $\exp l_x$ instead of $p(x) = \frac{\exp l_x}{\sum_{x'} \exp l_{x'}}$), as it turns out to be unnecessary for ensuring correctness.

$$\sum_x A_{x,z} = \pi_\theta(z), \qquad c_x = \frac{\exp l_x}{\sum_z A_{x,z}}, \qquad c_* = \max_x c_x, \qquad d_z = \frac{\sum_x A_{x,z} c_x}{\pi_\theta(z)},$$

$$\pi_\theta(x|z, l) = \frac{c_x}{c_*} \cdot \frac{A_{x,z}}{\pi_\theta(z)} + \left(1 - \frac{d_z}{c_*}\right) \frac{\exp l_x}{\sum_{x'} \exp l_{x'}}.$$

We now show that defining $\pi_\theta(x|z, l)$ in this way produces samples from the appropriate distribution.

**Proposition 6.** *$\pi_\theta(z)\pi_\theta(x|z, l)$ has the correct marginals, i.e.*

$$\sum_z \pi_\theta(z)\pi_\theta(x|z, l) = \frac{\exp l_x}{\sum_{x'} \exp l_{x'}}. \tag{30}$$

*Proof.* Expanding the right hand side,

$$\sum_z \pi_\theta(z)\pi_\theta(x|z,l) = \sum_z \pi_\theta(z)\left[\frac{c_x}{c_*}\cdot\frac{A_{x,z}}{\pi_\theta(z)} + \left(1 - \frac{d_z}{c_*}\right)\frac{\exp l_x}{\sum_{x'}\exp l_{x'}}\right]$$

$$= \frac{\exp l_x}{\sum_{x'}\exp l_{x'}} + \frac{1}{c_*}\sum_z \pi_\theta(z)\left[c_x\cdot\frac{A_{x,z}}{\pi_\theta(z)} - d_z\cdot\frac{\exp l_x}{\sum_{x'}\exp l_{x'}}\right].$$

Consider the quantity

$$V_x = \sum_z \pi_\theta(z)\left[c_x\cdot\frac{A_{x,z}}{\pi_\theta(z)} - d_z\cdot\frac{\exp l_x}{\sum_{x'}\exp l_{x'}}\right].$$

We see that

$$V_x = \sum_z \pi_\theta(z)c_x\frac{A_{x,z}}{\pi_\theta(z)} - \sum_z \pi_\theta(z)\frac{\sum_{x'}A_{x',z}c_{x'}}{\pi_\theta(z)}\frac{\exp l_x}{\sum_{x'}\exp l_{x'}}$$

$$= \sum_z c_x A_{x,z} - \sum_z\left(\sum_{x'}A_{x',z}c_{x'}\right)\frac{\exp l_x}{\sum_{x'}\exp l_{x'}}$$

$$= \sum_z \frac{\exp l_x}{\sum_z A_{x,z}}A_{x,z} - \sum_z\left(\sum_{x'}A_{x',z}\frac{\exp l_{x'}}{\sum_z A_{x',z}}\right)\frac{\exp l_x}{\sum_{x'}\exp l_{x'}}$$

$$= \frac{\sum_z A_{x,z}}{\sum_z A_{x,z}}\exp l_x - \sum_{x'}\frac{\sum_z A_{x',z}}{\sum_z A_{x',z}}\exp l_{x'}\frac{\exp l_x}{\sum_{x'}\exp l_{x'}}$$

$$= \exp l_x - \exp l_x\frac{\sum_{x'}\exp l_{x'}}{\sum_{x'}\exp l_{x'}} = 0.$$

We conclude that

$$\sum_z \pi_\theta(z)\pi_\theta(x|z,l) = \frac{\exp l_x}{\sum_{x'}\exp l_{x'}} + \frac{1}{c_*}V_x = \frac{\exp l_x}{\sum_{x'}\exp l_{x'}},$$

as desired. $\square$

# I  Experiments details

In all of our experiments we parameterize the two gadgets using multilayer perceptrons with two hidden layers of size 1024. Gadget 1 containes two sets of parameters, one for $\pi_{x,z}(p)$ and the other for $\pi_{y,z}(q)$. In gadget 2, for the last layer, we hold $\pi_\theta(Z)$ fixed as a uniform distribution over 20 latent causes. We used the Adam optimizer [Kingma and Ba, 2014] on the loss $g$. During training, we fix the softmax relaxation temperature to 1.

The gadgets were implemented in Jax and PyTorch frameworks [Bradbury et al., 2018, Paszke et al., 2019] and were trained on Nvidia GeForce RTX 2080 GPU.

**Minimizing variance over random logits**   For the first part of the experiment, we started by tuning learning rate for each gadget between 1e-5 and 1e-2 in six steps, training for 5000 iterations using a single random seed. We then selected the best learning rate for each condition ("independent" and "mirrored"), and repeated training with this learning rate for five additional random seeds, training for 50,000 iterations. We computed $g_{l^{(1)},l^{(2)}}(x,y) = (x-y)^2$ over random pairs $(p,q)$ in batches of 64 pairs and 16 drawn samples $(x,y)$ for each pair. We then evaluated on a set of 10,000 hold-out pairs $(p,q)$ for each random seed, and reported the standard deviation across the five seeds. (For the non-learned baselines, which are deterministic, this variance is instead over five independent evaluation sets.)

For the second part of the experiment, we trained the gadgets for 48,000 steps. We then evaluated on 10 different reward realizations while we trained the gadgets from scratch in each trial, averaging variance across 2,000 samples per pair.

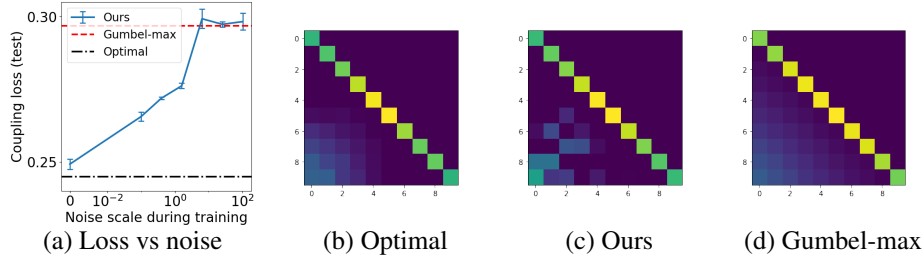

(a) Loss vs noise       (b) Optimal       (c) Ours       (d) Gumbel-max

Figure 5: Maximal coupling experiments with additional visualization. (a) When the training distribution is focused, we learn a near-maximal coupling. As the distribution becomes more diffuse, the learned coupling reverts to Gumbel-max. (b) Maximal coupling. (c) Our learned coupling in low-noise setting (Gadget 2). (d) Gumbel-max coupling.

**MDP counterfactual treatment effects** We learn the parameters of an MDP by interacting with the sepsis simulator for 10,000 times across all possible states-actions. Then, the behavior policy was trained over this MDP using policy iteration algorithm [Sutton et al., 2017], with full access to all MDP variables including a diabetes flag and glucose levels. The diabetes is present with a 20% probability, increasing the likelihood of fluctuating glucose levels. The patient is considered dead if at least three of his vital signs are abnormal, and discharged if all of his vital signs are normal. We set the rewards for these absorbing states to be -4 and 4 respectively.

Using the trained behavior policy, we draw 20,000 patient trajectories from the simulator with a maximum of 20 time steps, where the states are projected into a reduced state space of 6 discrete variables (without diabetes and glucose level). In total, there are 146 categories, including death and discharge states. Based on them, we learn the parameters of an additional MDP, and train the target policy over this MDP. In order to get $p(s'|s, a_{doctor})$, we looked at the projected MDP at the observed state-action pair. In order to get $q = p(s'|s, a_{intervention})$, we chose $a_{intervention}$ to be the argmax over the action space of the target policy.