# OpenReview forum: "Learning Generalized Gumbel-max Causal Mechanisms"
_NeurIPS.cc/2021/Conference — NeurIPS 2021 Spotlight_

### Official Review · Reviewer_g3MG · 2021-07-08

**Rating:** 7
**Confidence:** 3

**Summary:**

This paper introduces a continuously parameterized family of causal mechanisms where all members of the family are identical when used in a level 2 context but different when used in a level 3 context. The families contain the existing Gumbel-max technique proposed by Oberst & Sontag (2019), but a wide variety of other mechanisms can be learned by using gradient-based optimization. They show that the causal mechanisms can be learnt using a variant of Gumbel-softmax relaxation and that the mechanisms improve over Gumbel-max and other fixed mechanisms. Further, they show that the learnt mechanism is generalizable from a training set of observed outcomes and counterfactual queries to a testing set of observed outcomes and counterfactual queries.


**Limitations And Societal Impact:**

Yes, they do.

**Main Review:**

Section 1 & 2 & 3. The causal problem is not defined properly. If you are trying to solve the problem of learning causal mechanisms, please define mathematically what is a causal mechanism, what data you learn it from, what is the key challenge in general. I guess you don’t have to answer (not solve) these general questions with the Gumbel-max trick.  After presenting the general causal problem clearly, it will be better to explain the following two issues mathematically or using some toy examples of structural causal models (SCMs): (1) the non-identifiability of level 3 from level 2; (2) the requirement of counterfactuals (mentioned in lines 34-35); (3) Why we need to solve the problem using the family of causal mechanisms that generalizes the existing Gumbel-max SCMs. In the current introduction, Section 2 (Background) and Section 3 (Problem statement: Learning Implicit Coupling), I don’t see a well-defined causal inference problem. I feel like this is a coupling learning paper rather than a causal inference paper.

Overall, I suggest addressing the following three points mathematically: (1) formulate the problem of learning causal mechanisms; (2) formulate the requirement of counterfactuals; (3) highlight the advantage of learning specifically-tuned causal mechanisms.  I find the sections are well written, but the explanation for counterfactual stability in lines 88-94 is difficult to understand.

Section 4. Why do we need maximal couplings? I guess the answer is on lines 64-69. It looks like the variance of the objective function you will optimize to learn the coupling. How does this variance translate into the variance of the learnt coupling or causal mechanism? For the noise variable u, distributions p and q, could you define their space clearly?

Section 5. I am not very familiar with Gumbel-max and coupling theory. I must confess I don’t completely follow the discussion in this section. It is difficult for me to judge the novelty of the proposed method. Could you point out the novelties more clearly and explain why the continuously parameterized family of causal mechanisms you propose generalizes Gumbel-max?

Section 6. The related works on maximal coupling look good to me. I hope I know it earlier when I am reading through the paper.

Section 7. In Figure 3, you compare the variances of the estimates. Could you also compare the errors of the estimates?

Summary:
Overall, I find this paper interesting but a bit beyond my expertise. I lower my confidence level and will adjust my score appropriately after understanding the paper better by reading through your feedback and other reviews' comments.

Other comments:

In lines 38-39. Is the variance of an estimator a computational behaviour or statistical property (behaviour)?

Line 57:
You can always draw a sample from a categorical distribution.  What special about the Gumbel-max trick is that you can draw a differentiable sample from the continuous relaxation of a categorical distribution.

Line 81:
“that map states, actions, and noise” should be “that map actions, states and noise”

Equation (5) under line 115, a typo: “min. \theta”?




**Time Spent Reviewing:**

6 hours

---

> ### Author Response · Authors · 2021-08-07
> **Response and clarifications to reviewer g3MG**
>
> We thank the reviewer for their suggestions on how to improve the clarity of the introduction and background sections. We see this paper as being a synthesis of both causal inference and learning couplings, and hope to encourage interaction between the communities involved in each. As in Oberst and Sontag (2019), we consider the general problem of constructing a SCM that allows counterfactual (level 3) reasoning, when given access to all interventional (level 2) distributions. The key challenge is that there are often many different SCMs compatible with a set of level 2 distributions. Our idea is to circumvent this non-identifiability by finding an SCM with "good" counterfactual distributions (which can then be used to estimate treatment effects or other quantities of interest) based on interventional data, where a "good" counterfactual distribution is one with minimal loss according to some particular loss function, and the approach we take is to extend Gumbel-max SCMs with learned couplings. While this may be a non-standard causal inference problem, we believe it is a properly defined one, and would be happy to discuss this further if there are remaining concerns.
>
> Here we follow the reviewer's suggestion and state the causal components mathematically.  {\bf (1) formulate the causal mechanisms:} A causal mechanism is a function $f: R^d \times R^K \rightarrow \{1,...,k\}$  that maps noise u and logits l_1,...,l_k to a sampled outcome. Level 2 causal information considers interventions, i.e., probabilities and expectations, which in our setting is represented by the average treatment effect (ATE), $E_{u} R(f(u,l^{(1)}) - E_{u} R(f(u,l^{(2)})$ between a behavior policy $l^{(1)}$, e.g., a policy that is used in medical treatment, and an alternative policy $l^{(2)}$. We cannot generally identify a causal mechanism by intervening in real-world systems (i.e., using level 2 information). {\bf (2)  formulate the requirement of counterfactuals:} Oberst and Sontag (2019) proposed Gumbel-max causal mechanisms. The Gumbel-max causal mechanism sets $f(u,l) = \arg \max_{i \in \{1,...,K\}}  [l_i + u_i]$ where u_i are samples of standard Gumbel noise. Using this notation, the Oberst-Sontag ATE is computed by the argmax choice over the behavior policy and the counterfactual sample from the alternative policy  (see also lines 72-73 in our submission, setting h_1 and h_2 to R). {\bf (3) highlight the advantage of learning specifically-tuned causal mechanisms:} Our work suggests to use counterfactual information (level 3), cf. Section 5, to learn a function $f_\theta(u,l)$ that minimizes the variance of the treatment effect $R(f_\theta(u,l^{(1)})-R(f_\theta(u,l^{(2)})$. This gives a generalization of the Oberst-Sontag causal model that is more efficient for estimating ATE (specifically, when the variance of the counterfactual estimates are smaller, then fewer samples are required for good estimates).
>
> Section 4. We discuss maximal couplings because they are well known in the couplings literature, and because they provide a point of comparison for couplings that are based on SCMs. We do not require maximal couplings in our approach, and in fact maximal couplings are not always possible to obtain in a SCM (Proposition 2). Regarding the objective, the variance $Var_{(x,y) \sim \pi} [h_1(x)-h_2(y)]$ of the difference in outcomes under the coupling $\pi$ is itself the objective we seek to minimize, where we set $h_1$ and $h_2$ to be the treatment effects under (coupled) counterfactual outcomes x and y. (We apologize if calling $h_1$ and $h_2$ "cost functions" was confusing in this regard.) In Proposition 2, u, p, and q are allowed to be in arbitrary spaces, but in practice u is generally a vector of uniformly distributed random numbers and p and q are distributions over K discrete outcomes for some K.
>
> Section 5: To our knowledge we are the first to propose a parameterized family of SCMs based on couplings between counterfactual distributions, and the first to consider learning a particular SCM based on a loss function defined on those couplings. We build on Gumbel-Max SCMs, which have a single fixed coupling (the Gumbel-Max coupling) between their counterfactual distributions and have no parameters. Our parameterized family generalized Gumbel-Max in the sense that there exists a set of parameters for each of our gadgets that makes our SCM is the Gumbel-Max SCM: for gadget 1, we recover Gumbel-Max by setting $\pi_{x,z}(p) = p(x)\text{ if }x = z\text{ else }0$, and for gadget 2, we recover Gumbel-Max by setting $\pi(x | z, p) = p(x)$. We will include a derivation of this in our next revision.
>
> Section 7: We note that the average treatment effect is independent of the coupling used to estimate it, so all of our estimates are unbiased. As such, comparing (squared) errors is equivalent to comparing the variances of the estimator, up to a constant factor. (We also note that our error bars quantify the error in the measurements of the variance of our estimates.)
>
> Lines 38-39. Indeed, the variance of an estimator is a statistical property. We will fix this in the next revision.
>
> Line 57. While it is true that the Gumbel-Softmax trick allows you to draw differentiable samples from a continuous relaxation, the Gumbel-Max trick described in equation (2) serves a different purpose. It operates on the (discrete) argmax, and is useful because it allows us to draw coupled samples from two categorical distributions by using the same random noise.
>
> We would like to thank the reviewer for taking the time to write this thorough review and for providing valuable feedback despite the stated lack of expertise. We hope this response clarifies our work, and would be happy to answer any remaining questions.

---

> > ### Comment · Reviewer_g3MG · 2021-08-18
> > **Response**
> >
> > Thank you for clarifying the causal component, technical novelty and some experimental detail. I increase my score from 6 to 7. Overall, I find the paper interesting and novel. More works (e.g. providing preliminary sections in Appendix or some references) in revision will encourage better interactions between the causal and coupling communities.

---

> > > ### Author Response · Authors · 2021-08-26
> > > **Response to Reviewer g3MG**
> > >
> > > Thank you for engaging in discussion with us! This discussion has helped us understand how to improve the exposition of the paper, and we indeed hope it can lead to better interactions between the two communities. We have added an additional comment describing the specific changes we intend to make to the paper, and would welcome any additional feedback or suggestions you have.

---

### Official Review · Reviewer_4cdP · 2021-07-16

**Rating:** 7
**Confidence:** 3

**Summary:**

This paper generalizes the treatment effect estimation, in terms counterfactual and observed policies, of Oberst and Sontag (2019), to couplings. They build upon the Gumbel-max SCMs from Oberst and Sontag and show that with their proposed Gumbel-max coupling, they can learns causal mechanisms that minimize the counterfactual effect variance.

**Limitations And Societal Impact:**

I think the impact of this paper can be improved by providing a better intuition about the relation between couplings and counterfactual effect estimation, especially about why couplings lead to more desirable causal mechanisms. Moreover, as said before, improving the overall storyline could make the impact of this paper stronger.


**Main Review:**

I think this paper is quite well written for an audience that has read the paper of Oberst and Sontag (2019). Since I didn't read that paper beforehand, I had no clue what problem the authors were trying to solve. For example, the authors state in the introduction "Casting the problem this way ...", without explaining what the actual problem is, or describing the problem setting there in more detail. This was quite confusing, because the authors talk later about a data distribution that determines observed outcomes and counterfactuals. However, in practice, one has almost never access to those distributions IMO. Once I arrived at the Experiments section, I saw that the authors use simulations to get access to the counterfactual distributions. I think this confusion can be easily tackled by making the actual problem statement clear from the beginning.

Although the paper is well organized and each section is written very clearly, the overall storyline of the paper can be improved IMO. I think that the different sections (and concepts) are too loosely connected. For example, clarifying how the SCM for MDPs are related to the problem that the authors are trying to solve should be very clear from the beginning onward instead of being clarified (slightly) in the Experiments section. Or, on line 12 (and also on line 45) the authors state "We propose a parametrized family of causal mechanisms that generalize Gumbel-max", without showing this in the paper, or at least. I couldn't find it. Are the authors not generalizing the treatment effect instead of the causal mechanisms? This is a bit confusing, after reading what the definition of a causal mechanism of an SCM is.

The actual contributions of this paper could IMO be emphasized more. For example, one of the main contributions of this paper, that is IMO, that seeing the average treatment effect for a counterfactual vs. an observed policy in terms of couplings, could be emphasized more. This contribution is, for example, a bit downplayed by stating (in line 70) that [Maddison et al., Oberst and Sontag] exploit Gumbel-max couplings, which I think they never showed, since the proof of this is actually in the Appendix of this paper and not theirs. Or, in the introduction the authors state the questions about what mechanisms leads to the lowest variance estimates of treatment effects, which they didn't put much emphasis on in the paper, except by stating it somewhere in the related work section. Why not highlight this more and state how optimality and minimizing variance of the treatment effects are connected explicitly. I think these things that motivate this work should really be emphasized and pop out. This can make this paper easier to read and more strong.

At last, I was wondering about how to interpret the main objective? By generalizing the objective from Oberst and Sontag (2019) from difference between the outcomes to some (different) coupling between the outcomes, doesn't that also change the definition of the average treatment effect? That is, can we still talk about average treatment effect in some way? What is the intuition that lays behind using couplings instead?

Although the storyline could be improved, the rest of the paper is quite well written. Under the condition that all the proofs are correct (which I didn't check), I believe that this could potentially be a valuable contribution to NeurIPS.

Minor comments:
 - line 126 "... approach to the Gumbel-max couplings underlying Gumbel-max SCMs": Are they really underlying Gumbel-max SCMs?
 - line 134 "... Gumbel-max couplings are counterfactually stable ...": Where is this shown?
 - line 75: "casual" -> "causal"?
 - Figure 3: What is CRN? Please explain this somewhere.

Update: I updated my score.

**Time Spent Reviewing:**

6h

---

> ### Author Response · Authors · 2021-08-07
> **Response and clarifications to reviewer 4cdP**
>
> We thank the reviewer for their insightful suggestions and we are happy that they found the paper clear and well written. We assume access only to the interventional distribution by estimating an MDP using observations. Therefore, the average treatment effect, as defined by Oberst and Sontag, remains the same for any causal mechanism we construct in Section 5.  Following Oberst and Sontag settings, we use the simulator only for obtaining the initial observed trajectories and we don’t assume access to the simulator to get counterfactual probabilities. We will make this point clearer. We will also add a clarification on how our gadgets generalize the Gumbel-max causal mechanism
>
>
> Generalizing the causal mechanism does not change the definition of the treatment effect $R(f(u,l^{(1)}) -  R(f(u,l^{(2)})$ or its expectation $E_{u} R(f(u,l^{(1)}) - E_{u} R(f(u,l^{(2)})$  (the ATE), but it does change its variance. This allows us to fairly compare multiple causal mechanisms with regard to the variance of the treatment effect.
>
>
> Thanks for the suggestion to play-up the idea of viewing counterfactual treatment effects through the lens of coupling.

---

> > ### Comment · Reviewer_4cdP · 2021-08-26
> > **Response**
> >
> > Thank you for providing clarifications to my questions. However, one thing is not yet clear to me. In what sense are the causal mechanisms generalized? As I understand it now, the class of causal mechanisms are just a class of causal mechanisms of a specific form, nothing more. I thought that the main contribution of this paper is that by using couplings instead one learns these causal mechanisms that also minimize the counterfactual effect variance.

---

> > > ### Author Response · Authors · 2021-08-27
> > > **Additional clarifications**
> > >
> > > By "generalized" we mean that our class of causal mechanisms includes the Gumbel-max causal mechanism described by Oberst & Sontag (2019).
> > >
> > > More precisely, the Oberst-Sontag Gumbel-max causal mechanism sets $f(u, l) = \arg \max\_{i \in \{1, \dots ,K\}}  [l\_i + u\_i]$ where $u\_i$ are samples of standard Gumbel noise. Our starting point is the Average Treatment Effect (ATE) in the Gumbel-max casual model $E\_{u} [R(f(u,l^{(1)})) - R(f(u,l^{(2)}))] = E\_{u} [R(f(u,l^{(1)}))] - E\_{u} [R(f(u,l^{(2)}))]$. Our work suggests using counterfactual information to learn a function $f\_\theta(u,l)$ that minimizes the variance of the treatment effect $R(f\_\theta(u,l^{(1)}))-R(f\_\theta(u,l^{(2)}))$. The shared exogenous random variables $u$ create a coupling of the two policies $l^{(1)}$ and $l^{(2)}$ by producing coupled samples $(f\_\theta(u,l^{(1)}), f\_\theta(u,l^{(2)}))$ and we learn the parameters $\theta$ to minimize the variance $\text{Var}\_{u}[R(f\_\theta(u,l^{(1)}))-R(f\_\theta(u,l^{(2)}))]$. The sense in which this is a generalization is that there is a setting of $\theta$ that causes $f\_\theta$ to reduce to the Gumbel-max mechanism. We show in the paper that learning $\theta$ yields a causal model that is more efficient for estimating ATE (when the variance of the counterfactual estimates are smaller, then fewer samples are required for good estimates). Our point in our previous response was that the definition of the treatment effect remains the same after substituting $f\_\theta(u, l)$ in place of $f(u, l)$, and moreover every causal mechanism in our parameterized family has the same value of the ATE since the ATE depends only on the policies $l^{(1)}$ and $l^{(2)}$ and not on the way they are coupled.
> > >
> > > While our SCMs are a generalization of the Gumbel-max SCM, they are still ordinary SCMs. We are not generalizing the notion of SCM itself or attempting to express every SCM, and it is true that our family only includes SCMs of a particular form as the reviewer notes.

---

> > > > ### Comment · Reviewer_4cdP · 2021-08-30
> > > > **Response**
> > > >
> > > > Thank you for answering my question and making clear that you are actually generalizing the causal mechanisms of a Gumbel-max SCM to more flexible causal mechanisms. I increase my score from a 6 to a 7. I find this paper interesting and novel, but I hope that the authors can clarify most of the questions raised by the different reviewers in the main text of the revised version.

---

### Official Review · Reviewer_cSRr · 2021-07-16

**Rating:** 7
**Confidence:** 2

**Summary:**

The authors propose a method to parametrize the functions of a certain class of structural causal models (SCM) called Generalized Gumbel-max SCMs. The aim is to train the model while minimizing the variance of treatment (counterfactual?) effects computed from them.
The paper introduces two different families of functions (gadgets) that can be learned through an optimization process aiming to find a maximal coupling of a pair of logits vectors defining a discrete distributions.
The experimental section compares the two gadgets with the Gumbel-max representation from Oberst and Sontag, and a cumulative distribution functions (CDF) inversion method. The experimental results seem to suggest that the proposed gadgets improve may achieve lower variance.

**Limitations And Societal Impact:**

There is discussion on technical limitations.
There is no negative societal impact that requires discussion at this point.

**Main Review:**

I am not familiar reparametrization or couplings, but as far as I can tell, their application to causal modeling tasks is novel. It seems that relevant work as been cited (Oberst and Sontag, 2019) and the differences have been addressed.

I am unable to assess the soundness of most of the technical results as they are outside my are of expertise. However, I do have some questions regarding the specifics of the causal task that is attempted which I will discuss shortly. Overall, the paper does seem to support the claims and the experiments provided seem to support the benefits from employing the proposed approach.

In terms of clarity it is hard for me to understand what exactly is the data (observational, interventional) assumed to be known for the task. Also, what is exactly the relationship between the logits vectors and that data? Does this approach work for any type of input data or are you assuming a particular form? I believe a simple running example could help ground these aspects.
The introduction touches upon the different levels of the ladder of causation and the fact that it is not possible, in general, to infer level 3 data solely from level 2 data. Nevertheless, it is not clear what are the level 3 quantities that the authors aim to estimate with minimal variance.
It is not clear to me what kind of counterfactual is being evaluated. Also, the motivating observation "if we only care about our models being consistent with interventional data, then we have flexibility in the choice of causal mechanism" seems to disregard the level 3-level 2 impossibility mentioned in the introduction. I hope the authors could clarify these points in their response.

In terms of significance, I am unable to assess the applicability of the proposed method, mainly to the lack of a clear picture of the input data, the query of interest and the assumptions being made about the true underlying data generating process. It is very likely that this picture could be understood from the paper but I believe it should be more explicit and easy to grasp for the average reader trained in causal inference. On a related note and in contrast with Oberst and Sontag, 2019, the SCM formulation in the paper seems to assume the absence of latent confounding, is this the case?

## Update
After reading the response from the authors, I feel they have clarified my main concerns. I'm updating my score.

**Time Spent Reviewing:**

7

---

> ### Author Response · Authors · 2021-08-07
> **Response and clarifications to reviewer cSRr**
>
> We are happy the reviewer found our work novel and that the paper supports its claims and the experiments support the benefits from employing our method.
>
> We would like to clarify that the average treatment effect (ATE) itself (which we seek to estimate in our experiments) is a level 2 quantity, depending only on the interventional distributions and not on their counterfactual relationship. However, the variance of the treatment effect is a level 3 quantity, since it depends on the pair of outcomes under the original and alternative policies. This is what we mean by our motivating observation: if our ultimate goal is to estimate a level 2 quantity (the ATE), we are free to choose an arbitrary level 3 causal mechanism that is consistent with the interventional distributions of interest. In particular, we can choose the level 3 mechanism whose variance is smallest, leading to a more accurate estimate of the ATE.
>
> The key problem we’re tackling is that there are many causal mechanisms (or reparameterization tricks) that are consistent with a given distribution $p_i = e^{l_i}$. That is, there are many deterministic mappings from a noise source u and logits $l_1,...,l_K$  that yields samples from $p_i = e^{l_i}$. Our approach is about how to choose a causal mechanism for the given distribution that has desirable properties (specifically, minimal variance) when used in a counterfactual setting.
>
> In the following we give a simple running example, in a restricted setting of a single choice, over two distributions whose logits are $l^{(1)}$ and $l^{(2)}$. The Gumbel-max causal mechanism $f(u,l)$, as defined by Oberst and Sontag (2019), sets $f(u,l) = \arg \max_{i \in \{1,...,K\}}  [l_i + u_i]$ where u_i are samples of standard Gumbel noise. Our starting point is the Average Treatment Effect (ATE) in the Gumbel-max casual model $E_{u} R(f(u,l^{(1)}) - E_{u} R(f(u,l^{(2)})$ between a behavior policy $l^{(1)}$, e.g., a policy that is used in medical treatment, and an alternative policy $l^{(2)}$. The ATE is a level 2 quantity since it considers interventional causal information, namely, the probability of an intervention (i.e., expectation of a causal choice over the random choice u, the exogenous variables). Using this notation, the Oberst-Sontag ATE is $E_{u} [R(f(u,l^{(1)}) - R(f(u,l^{(2)})]$. Our work suggests to use counterfactual information (level 3), cf. Section 5, to learn a function $f_\theta(u,l)$ that minimizes the variance of the treatment effect $R(f_\theta(u,l^{(1)})-R(f_\theta(u,l^{(2)})$. When we minimize the treatment effect’s variance (while fixing the interventional log-probabilities $l^{(1)},l^{(2)}$, there is a degree of freedom, which turns to be the maximization of the correlation $E_u [R(f_\theta(u,l^{(1)}) R(f_\theta(u,l^{(2)})$). This correlation is based on the counterfactual reasoning about $R(f_\theta(u,l^{(2)}))$ given the exogenous choice of $R(f_\theta(u,l^{(1)}))$, which we obtain by coupling (see also lines 72-73 in our submission, setting h_1 and h_2 to R). We compare to the Oberst-Sontag causal model and show that the variance of the treatment effect of our learned function $f_\theta(u,l)$ is significantly lower than the variance of the Oberst-Sontag causal model (when the variance of the counterfactual estimates are smaller, then fewer samples are required for good estimates). Our experimental setup in Section 7.3 follows the experimental setup of Oberst and Sontag and we outperform their method in their setup.
>
> We use the Oberst and Sontag experimental setup in Section 7.3 and their data.
> Following Oberst and Sontag, we assume to have observations of the form (a, s, s’). By learning an MDP we have access to the interventional probabilities  p(s’|s,a) . Our approach works with any two categorical distributions and observed outcomes from one of the distributions.
>
> Here are answers to concerns raised by the reviewer:
>
> > “what exactly is the data (observational, interventional) assumed to be known for the task?”
>
> We assume access to level 2 quantities (information that can be acquired via interventions). Specifically, we assume we observe the average treatment effect of the behavior policy.
>
> > “Also, what is exactly the relationship between the logits vectors and that data?”
>
> In Section 7.3 we evaluate our method on the Sepsis model that was used by Oberst and Sontag. The logits vectors represent transition probabilities log p(s’ | s, a). These transition probabilities are estimated from the behavior policy, i.e., from the observed data of medical treatment.
>
> > “Does this approach work for any type of input data or are you assuming a particular form?”
>
> The general approach is applicable to any SCM causal mechanism with discrete outcomes.
>
> > “Nevertheless, it is not clear what are the level 3 quantities that the authors aim to estimate with minimal variance”
>
> Thank you for emphasizing this point: The ATE is actually a level 2 quantity, and all choices of causal mechanism yield the same ATE. The *variance* of the counterfactual estimate changes based on the causal mechanism, though, and this is our key observation: choosing the causal mechanism that yields minimum variance is an interesting criterion for choosing amongst causal mechanisms.
>
> > “"if we only care about our models being consistent with interventional data, then we have flexibility in the choice of causal mechanism" seems to disregard the level 3-level 2 impossibility mentioned in the introduction”
>
> ATE is determined by interventional data, i.e., it considers expectation after an intervention and therefore it is a level 2 quantity. But the variance of the counterfactual ATE estimate depends on the causal mechanism. If we only cared about choosing a causal mechanism that produced the correct ATE, then there’s no reason to prefer one causal mechanism versus another (and this is related to the non-identifiability). But if we instead say that we want a causal mechanism that gives minimum ATE variance, then this gives a criterion for choosing one causal mechanism over another, even though they both produce the same estimates of ATE.
>
> > “assumptions being made about the true underlying data generating process”
>
> We’re only assuming access to quantities that can be measured via intervention (level 2), i.e., access to marginal distributions and expectation (average treatment effect).
>
> > “On a related note and in contrast with Oberst and Sontag, 2019, the SCM formulation in the paper seems to assume the absence of latent confounding, is this the case? “
>
> Our approach should be applicable in the same set of cases that Obert & Sontag is applicable. While this is implicit in their work, both papers are about criteria for choosing a level 3 model that is consistent with level 2 information. These questions are more about the causal mechanisms governing individual variables (the $f: R^d \times R^K \rightarrow \{1,...,k\}$  that maps noise u and logits l_1,...,l_k to a sampled outcome) than the surrounding DAG structure.
>
> > “I believe it should be more explicit and easy to grasp for the average reader trained in causal inference”
>
> Thanks for the feedback. This paper sits in between causal inference, couplings, and reparameterization tricks. We tried to write it in a way that would be accessible to all three audiences, but we will take the reviewers feedback to improve the exposition for the readers trained in causal inference.

---

> > ### Comment · Reviewer_cSRr · 2021-08-22
> > **Re: response**
> >
> > Thank you for answering my questions and providing clarification. I increased my score to 7. I hope you can include some version of the running example or other in the main body of the paper to make things concrete for the readers.

---

> > > ### Author Response · Authors · 2021-08-26
> > > **Response to Reviewer cSRr**
> > >
> > > Thank you for engaging in discussion with us! We will be sure to include the running example in the main body of the paper as you suggest. We have added an additional comment describing the specific changes we intend to make to the paper, and would welcome any additional feedback or suggestions you have.

---

### Official Review · Reviewer_eGhr · 2021-07-17

**Rating:** 5
**Confidence:** 2

**Summary:**

The authors propose a method extending Gumbel SCM for doing counterfactual reasoning.

**Limitations And Societal Impact:**

I don't think this work would have negative societal impact.

**Main Review:**

The paper could have been written a little more clearly in places. For example, Figure 2 did not appear to be linked back to the text. It would have been helpful to have a little more explanation. “Discriminative” was misspelled line 256.

For originality, I haven't seen any other papers like it, though perhaps other reviewers have.

The quality of the theory is clear.

This is significant insofar as counterfactual reasoning is a persistent problem for causal inference. It would have been helpful if the approach had been compared to other approaches for doing counterfactual reasoning. At least a discussion could have been included as to why this covers cases different from what, say, Pearl's methods would have covered.


**Time Spent Reviewing:**

1

---

> ### Author Response · Authors · 2021-08-07
> **Response and clarifications to reviewer eGhr**
>
> We are happy the reviewer appreciated the quality of the theory, the originality of our approach, and the significance of the problem. We take on board the feedback that we could have emphasized in greater detail the comparison to standard approaches. We provide some additional details here, which we will add to the next revision of the paper.
>
> Our work concerns the problem of choosing a causal mechanism. A causal mechanism is a function $f: R^d \times R^K \rightarrow \{1,...,k\}$  that maps noise u and logits $l_1,...,l_k$ to a sampled outcome. We cannot generally identify a causal mechanism by intervening in real-world systems (i.e., using level 2 information), but as the reviewer notes, counterfactual reasoning is an important problem.
>
> A well-known way to choose a causal model is to impose assumptions that make the causal model identifiable. For instance, Pearl (2000) shows that if outcomes are assumed to be monotonic with respect to the random noise, there is a unique way to specify the counterfactual distributions. This corresponds to our inverse-CDF condition in our experiments. Our approach does not require making these assumptions, and we show that in some cases a learned causal mechanism leads to better performance.
>
> More recently, Oberst and Sontag (2019) proposed Gumbel-max causal mechanisms. The Gumbel-max causal mechanism sets $f(u,l) = \arg \max_{i \in \{1,...,K\}}  [l_i + u_i]$ where u_i are samples of standard Gumbel noise. Our starting point is the Average Treatment Effect (ATE) in the Gumbel-max casual model $E_{u} R(f(u,l^{(1)}) - E_{u} R(f(u,l^{(2)})$ between a behavior policy $l^{(1)}$, e.g., a policy that is used in medical treatment, and an alternative policy $l^{(2)}$. The ATE is a level 2 quantity since it considers interventional causal information, namely, the probability of an intervention (i.e., expectation of a causal choice over the random choice u, the exogenous variables). Using this notation, the Oberst-Sontag ATE is $E_{u} [R(f(u,l^{(1)}) - R(f(u,l^{(2)})]$. Our work suggests to use counterfactual information (level 3), cf. Section 5, to learn a function $f_\theta(u,l)$ that minimizes the variance of the treatment effect $R(f_\theta(u,l^{(1)})-R(f_\theta(u,l^{(2)})$. This gives a generalization of the Oberst-Sontag causal model that is more efficient for estimating ATE (specifically, when the variance of the counterfactual estimates are smaller, then fewer samples are required for good estimates). We compare to the Oberst-Sontag causal model and show that the variance of the ATE of our learned function $f_\theta(u,l)$ is significantly lower than the variance of the Oberst-Sontag causal model. Our experimental setup in Section 7.3 follows the experimental setup of Oberst and Sontag and we outperform their method in their setup.
>
> Pearl, J. Probabilities of causation: three counterfactual interpretations and their identification. Synthese, 121 (1):93–149, 2000.

---

### Author Response · Authors · 2021-08-26
**Planned changes in next revision**

We would like to thank all of the reviewers for their feedback, which has been extremely valuable in helping us understand how we can improve the paper. We will put in sufficient time and energy in order to incorporate all of the excellent suggestions.

If the paper is accepted, there will be an extra page of allowed content, and we will use all of it to make the exposition clearer based on the discussions with reviewers. Here’s our plan. Any additional feedback is welcome.
- **Expand discussion on families of causal mechanisms.** We’ll bring forward the discussion around L79 of what a causal mechanism is and also include a discussion of the non-identifiability from level 2 information (suggestions from Rev g3MG). We will be more explicit that there is a family of causal mechanisms that are consistent with observed level 2 information, and we’ll foreshadow our approach by introducing a parameterized family of causal mechanisms that all have the same level 2 behavior.
- **Expand on Oberst & Sontag background.** We’ll then discuss previous approaches to dealing with the non-identifiability including Pearl (2000) (included in the response to Rev eGhr) and more background on Oberst and Sontag (2019), to address Rev 4cdP’s comment that it was necessary to read Oberst & Sontag first and Rev eGhr’s comment that an explicit discussion of the pros and cons compared to Pearl’s approach would have been appreciated. We will spend more time on explaining the counterfactual stability assumption from Oberst & Sontag. We agree the paper should be self-contained.
- **Distinguish level 2 and 3 quantities with an example.** Afterwards, we’ll bring in the running example from the discussion with Rev cSRr and discuss which quantities are level 2 (ATE) and which are level 3 (variance of ATE estimate), as in the discussions with Reviewers g3MG, cSRr, and eGhr. We’ll explain in the context of the example that our approach leverages a degree of freedom to optimize the level 3 quantity without changing the level 2 quantity. We’ll add a discussion about the implications of the assumptions made by each approach.
- **Add an example of input data.** We will add an explicit discussion of what the concrete input data looks like for an example application in Section 3 when stating the problem, including the relationship between logits in the problem description and the data in a real application, based on comments from Rev cSRr & 4cdP.
- We will add a discussion of when the approach is applicable in Sec 3 as well, following the response to Rev cSRr.
- We will add a paragraph in Sec 3 after “Relationship to 1-Wasserstein” that explicitly talks about how to interpret the problem statement in terms of causal inference, based on comments from Rev cSRr.
- In Sec 5, we will add an explicit description of how Gadget 2 generalizes Gumbel-max by showing that there is a simple setting of the learnable parameters that leads to exactly the Gumbel-max mechanism.
- We’ll refine the emphases according to Rev 4cdP’s suggestions in the paragraph starting “The actual contributions of this paper could IMO be emphasized more.” This will involve changes to the intro and also adding more discussion throughout the paper about the significance of the various results by drawing more comparisons against what the alternatives would be.
- We will fix the typo and other small issues raised by the reviewers.

Thank you all for the reviews. We believe this restructuring will make the paper stronger and more accessible. We’re grateful for your time and thoughtfulness.

---

### Decision · Program_Chairs · 2021-09-28

**Decision:**

Accept (Spotlight)

**Comment:**

This paper explores an understudied connection between causal inference and couplings learning, building upon very recent work from Oberst & Sontag. More specifically, it considers the general problem of constructing a SCM that allows counterfactual (level 3) reasoning, when given access to all interventional (level 2) distributions. The key challenge is that there are often many different SCMs compatible with a set of level 2 distributions. To circumvent this non-identifiability issue, they find a SCM with "good" counterfactual distributions based on interventional data, where a "good" counterfactual distribution is one with minimal loss according to some particular loss function. Here, the approach they take is to extend Gumbel-max SCMs with learned couplings.

While the reviewers initially had some concerns re. clarity/significance of the contribution, the authors' replies and subsequent discussion cleared up these concerns and a majority of reviewers agreed that it is worth publishing at neurips.

**Consistency Experiment:**

NeurIPS has a long history of experimentation. In 2014, NeurIPS ran an experiment in which 10% of submissions were reviewed by two independent committees to quantify the randomness in the review process. This year, we repeated a variant of this experiment to see how the quality of the review process has changed over time.  This paper was part of the experiment and was therefore assigned to two committees (consisting of reviewers, an Area Chair, and a Senior Area Chair) that reached independent decisions.  If both committees made the same recommendation, this recommendation was followed. If a single committee recommended acceptance, the paper was accepted (with the exception of a few cases in which the other committee identified what we considered a fatal flaw, e.g., an error in a key result).

This copy’s committee reached the following decision: **Accept (Spotlight)**

The other committee assigned to the paper recommended **Reject**.  You can find the other set of reviews, along with any follow up discussion with the authors here:
https://openreview.net/forum?id=oErdeq9ajjX